# Harnessing Spatial Dependency for Domain Generalization in Multivariate Time-series Sensor Data

## Abstract

Multivariate time-series (MTS) data from multiple sensors often vary across domains due to factors like sensor misalignment, reattachment, or individual differences, posing significant challenges for domain generalization (DG). Existing methods inadequately address the alignment of domain-specific spatial dependencies across different domains in MTS data, as they often assume a unified invariant spatial structure and overlook the distributional discrepancies arising from varying sensor relationships. To address this limitation, we propose ASAM (*Adaptive Spatial Dependency Alignment in MTS Data for Domain Generalization*), a novel framework that adaptively aligns spatial dependencies across domains. ASAM proposes a DG layer with domain generalization loss function and two-view regularization loss functions to align spatial dependencies between domains adaptively. We adopt a two-phase approach to align different sets of domains effectively. An input-aware graph generation process and a GNN-based DG layer, coupled with the domain generalization loss function, adaptively align the spatial dependencies learned in the second phase with those from the first phase, ensuring a more precise alignment. We additionally incorporate a two-view regularization method to effectively capture underlying spatiotemporal information comprised of spatial decorrelation loss and Gaussian kernel loss. Our theoretical analysis demonstrates that ASAM effectively assimilates information bottleneck, ensuring robustness across diverse distributions. Extensive evaluations of the four real-world datasets show ASAM outperforms ten recent baselines. To the best of our knowledge, this work is among the first to explore DG approaches for MTS data by focusing on spatial dependency alignment. Our code is available at https://anonymous.4open.science/r/ASAM.

## 1 Introduction

In the context of analyzing human-related behavior with multivariate time-series (MTS) data, it is essential to identify the complex interplay between multi-sensor structure and concurrent time-series features. There are several task-specific feature learning methods regarding inherent spatial and temporal aspects of MTS data to capture intricate spatiotemporal features of the data (Yi et al., 2024; Cai et al., 2024). Specifically, recent studies adopt a Graph Neural Network (GNN) structure to obtain structural information between sensors (Wang et al., 2024a;b). Yet, sensor-driven MTS data often display varying signal values under different conditions—such as data collected from different individuals or from the same person at different times with sensor reattachment (Farina et al., 2014; He et al., 2023), resulting in multiple separate domains related to multi-sensor (spatial) dependencies.

Domain generalization (DG) methods have recently attracted significant attention to mitigate the distribution shifts. DG methods are especially practical for the healthcare field where access to the target domains is rigorously restricted due to privacy concerns (Deng et al., 2024; Zhang et al., 2021). Applying previous feature-learning methodologies in the DG settings has certain constraints, as they primarily focus on learning spatial features within the observed domains without any consideration of domain information, as shown in Figure 1-(a). This approach reduces their effectiveness in addressing distribution shifts in unseen domains, such as the case of T1 in the figure. Moreover,

Figure 1: Comparison between two existing works and ours. It displays a multi-domain setting with S1, S2, and S3 as source domains for training and T1 as target domain for inference. Lowercase and uppercase letters beside the sensors represent learned sensor features with the given domain-specific information and domain-generalized information, respectively. The 'Module' learns spatial alignment information from domain-specific to domain-generalized spatial dependency.

most DG methods tend to overlook the discrepancy of spatial dependencies across domains. There are some studies besides the MTS sensor data that propose graph-based DG approaches (Miao et al., 2022; Sui et al., 2022) that account for spatial dependencies by extracting invariant graph structures. However, these methods remain limited in addressing the distributional discrepancies arising from distinct sensor dependencies for domains in MTS sensor data. As depicted in Figure 1-(b), existing methods that focus on extracting a domain-sharing uniform spatial structure may produce subpar performance on a target domain that exhibits significantly different sensor relations with the observed domains, as on T1 in the figure. Therefore, results may be inconsistent depending on how well the target spatial structure matches the invariant spatial structure captured in the source domain. To the best of our knowledge, DG approaches specifically tailored for MTS data, considering the discrepancy of spatial dependencies across domains, have not been extensively examined.

To address the limitations of previous research, we propose ASAM (*Adaptive Spatial Dependency Alignment in MTS Data for Domain Generalization*) by adaptively aligning spatial dependencies. As shown in Figure 1-(c), ASAM optimizes an adaptive spatial alignment module, transitioning from spatial dependencies with domain-specific features (a, b, c, d) to spatial dependencies with domain-generalized features (A, B, C, D), and this alignment module is robust against various spatial-wise distributions T1 that occur in MTS sensor data. For instance, in the target domain T1 shown in Figure 1, where both types of existing methods may struggle, the module of ASAM adaptively aligns domain-specific features, such as 'a' and 'b', with domain generalized features, 'C' and 'D', enabling effective generalization. ASAM proposes a domain generalization layer (DG layer) and a domain generalization loss for DG in MTS sensor data to align across numerous unseen domains. ASAM is optimized through a two-phase approach, with each phase involving distinct input-source data to facilitate spatial alignment, as the given source domain set (S1, S2, S3) is divided into two parts —S1 and S2 for the phase 1, and S3 for the phase 2 at random — as depicted in Figure 1. The initial phase comprehensively captures overall feature information with a spatiotemporal model, while the second phase focuses on learning inter-domain information by incorporating the DG layer and its loss function. Specifically, we propose a loss function designed to ensure that the embedding processed by the DG layer closely resembles the non-processed embedding. In addition, a two-view regularization loss is incorporated to enhance the capability of the alignment by regularizing to obtain underlying spatiotemporal features from the first phase. We theoretically demonstrate that our DG layer assimilates invariant information from the first phase data, thus confirming that the DG layer is robust on various distributions.

We rigorously evaluated our model using several real-world datasets, including sEMG and human activity recognition (HAR) datasets. Empirical analyses affirmed the model's superior efficacy compared to ten recent baselines.

## 2 RELATED WORK

### 2.1 DOMAIN GENERALIZATION

Domain generalization (DG) intends to close the gap between the source and target data distributions. Since the distribution shift problem can occur in various fields, including computer vi-

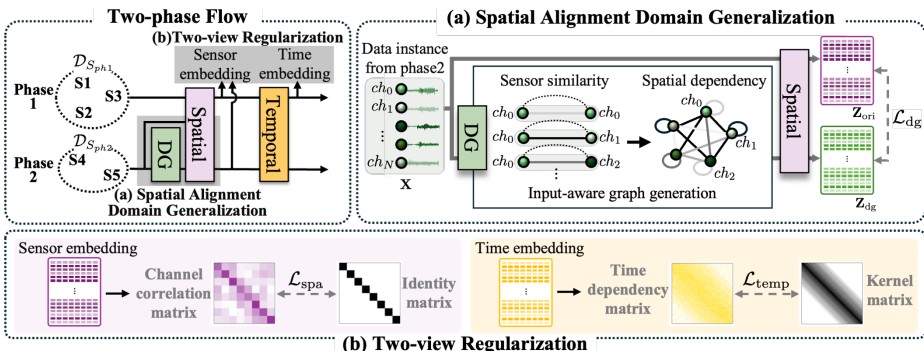

Figure 2: Illustration of the overall flow of ASAM. The upper left section of the figure illustrates the two-phase flow of ASAM, featuring S1, S2, ..., and S5 as the source domains. The proposed spatial alignment domain generalization, which incorporates the domain generalization layer and its associated loss function used in the second phase, is illustrated in Section (a). Section (b) presents the two-view regularization method for extracting intrinsic features from the spatiotemporal model.

sion (Thota & Leontidis, 2021), natural language processing (Glorot et al., 2011), and graphs (Miao et al., 2022; Fan et al., 2022), the attention of DG is actively increasing. Most DG approaches try to discern the invariant information, and it can be classified into two: preventing the main model from overfitting to the source data and learning domain invariant information. The first approach utilizes data manipulation (Wang et al., 2022) or incorporates extra learning like meta-learning (Qin et al., 2023) and self-supervised learning (Kim et al., 2021). The latter approach dissects the representation into domain-specific and domain invariant information (Miao et al., 2022).

Applying existing DG approaches to multivariate time-series data is challenging because of its complex correlations with spatial and temporal aspects. Previous works mainly focus on capturing time dependent factors (Deng et al., 2024; Hu et al., 2022). (Qian et al., 2021) utilizes DG for sensor-based human activity recognition based on the variational autoencoder (VAE) framework, incorporating two encoders to disentangle domain-agnostic and domain-specific representations. (Lu et al., 2022) proposes the DG method exploiting the local and global correlation among segments of a single time-series data to learn domain invariant temporal features. However, these approaches only focus on the temporal correlation, while spatial dependency across domains is also essential in the MTS domain. There are several graph-based DG studies (Miao et al., 2022; Sui et al., 2022; Fan et al., 2022) that employ causality-based approaches to identify non-causal substructures and isolate domain invariant graphs. However, these methods extract a single invariant spatial dependency that may be suboptimal in MTS sensor data that has various distinct distributions. ASAM, on the other hand, aligns the spatial dependency across domains, which is more robust on various distinct distribution settings like MTS sensor data.

## 2.2 SPATIAL ALIGNMENT IN DOMAIN ADAPTATION SETTING

Domain generalization (DG) and domain adaptation (DA) both intend to close the gap between the source and target data distribution. The main difference between them lies in the access to target data: domain adaptation leverages target data during training, whereas domain generalization relies exclusively on the source data without utilizing target data. Domain adaptation allows some information according to their problem setting, such as target domain data feature for unsupervised domain adaptation (UDA).

Even though there are some differences between UDA and domain generalization, there are some studies of UDA on MTS data that considers spatial information. Most methods (Lu & Sun, 2024) find the domain-invariant information similar to existing DG methods as described in Figure 1-(b). However, there is a study that similarly considers the spatial correlations (Wang et al., 2023). They typically adopt a fixed sensor matching approach to consider the sensor structure of MTS data (Wang et al., 2023). This assumption raises concerns in cases such as a left-handed user in the source domain and a right-handed user in the target domain. Our work can symmetrically align the sensor dependencies, addressing the potential limitations of (Wang et al., 2023).

# 3 METHODOLOGY

## 3.1 PROBLEM DEFINITION

Each instance within this dataset can be represented as $\mathbf{X} \in \mathbb{R}^{N \times T}$, where $N$ indicates the number of sensors (channels) and $T$ denotes the number of timestamps. The dataset can be formalized as $\mathcal{D} = \{(\mathcal{X}^{(i)}, \mathcal{Y}^{(i)})\}_{i=1}^{DOM}$, where $DOM$ indicates the number of domains and $\mathcal{X}^{(i)}, \mathcal{Y}^{(i)}$ are the data and label set of $i$-th domain respectively. The whole dataset $\mathcal{D}$ can be decomposed into source domain set $\mathcal{D}_S$ and target domain set $\mathcal{D}_T$. The set of target domains $\mathcal{D}_T$ is utilized in the inference stage and is non-accessible during the model learning process. In ASAM, the source domain set $\mathcal{D}_S$ is randomly divided into two subsets, $\mathcal{D}_{S_{ph1}}$ and $\mathcal{D}_{S_{ph2}}$, where $\mathcal{D}_{S_{ph1}}$ is the first phase domain set and $\mathcal{D}_{S_{ph2}}$ is the second phase domain set.

## 3.2 OVERALL FRAMEWORK

This section presents our proposed model on multivariate data. The overall structure of our model is depicted in Figure 2. As illustrated in the figure, our training process is divided into two phases. In the first phase, the spatiotemporal model $f_\theta$ is exclusively trained using $\mathcal{D}_{S_{ph1}}$, where $\theta$ is the parameter of the spatiotemporal model, which incorporates a spatial layer and a temporal layer. The spatial layer captures the spatial sensor relations in a single domain, and its representation is denoted as $\mathbf{Z} \in \mathbb{R}^{N \times T}$. The temporal layer then processes $\mathbf{Z}$ and obtains the holistic spatiotemporal representation $\mathbf{C} \in \mathbb{R}^{F \times T}$, where $F$ is the feature dimension of each temporal dimension. For the experiment, to validate the effectiveness of the proposed spatial alignment-based domain generalization, we adopt a simple spatiotemporal model composed of a GNN and LSTM for the spatial and temporal layers, respectively. The details of the spatiotemporal structure are provided in the Appendix. The second phase integrates the domain generalization layer (DG layer) $a_\pi$ before the spatiotemporal model $f_\theta$, constructing domain generalization model $g_{\pi,\theta}$, where $\pi$ is the parameter for the DG layer. The objective of the second phase is to adaptively align the spatial dependency of $\mathcal{D}_{S_{ph2}}$ to spatial structure extracted from the first phase domain $\mathcal{D}_{S_{ph1}}$. We process the second phase model in the inference stage to align unseen domain $\mathcal{D}_T$ into the source domains. The DG layer employs a GNN-based approach with input-driven adjacency matrix generation to adaptively address diverse spatial structures of unseen domains. A domain generalization loss function is applied in the second phase to align the spatial dependency across domains, enabling the domain generalization layer to learn input-aware spatial alignment. We further introduce a two-view regularization loss across both phases to effectively identify the spatial alignment by regularizing the spatiotemporal model to focus more on underlying features.

The following sections explain the proposed spatial alignment domain generalization method and two-view regularization. Furthermore, we will outline the final loss functions used to optimize our model and analyze the time complexity of ASAM. A theoretical analysis exhibits that the proposed loss functions lead to domain generalization for the MTS sensor data.

## 3.3 SPATIAL ALIGNMENT DOMAIN GENERALIZATION

A solution for adapting to various unseen distributions with consistent results is to adaptively align the unseen domain's spatial dependency to the trained spatial dependency. Since extracting invariant distribution with an assumption of shared spatial structure across domains may induce deficient results at inference, as we discussed in Figure. 1, we propose a **spatial alignment domain generalization method**. We address the distribution shift by adopting an additional domain generalization layer and loss function in the second phase which aligns the spatial dependency across domains.

A GNN-based domain generalization layer (DG layer) is proposed with input-driven adjacency matrix generation in which spatially related information is preserved for aligning distinct distributions. The input-driven graph is constructed solely with the given feature matrix designed to learn relations that are salient to the input features. Consequently, the edge weight between sensors with similar features is designed to be higher, while it is lower for those with dissimilar features. Therefore, the edge weight between sensors that is likely to be aligned is preferred to have high values and vice versa.

The similarity matrix $\mathbf{A}_{\text{dg}} \in \mathbb{R}^{N \times N}$ can be expressed as follows:

$$\mathbf{A}_{\text{dg}} = \sigma(\frac{(\mathbf{W}_1\mathbf{X})(\mathbf{W}_2\mathbf{X})^{\mathsf{T}}}{\sqrt{d_k}}), \tag{1}$$

where $\mathbf{W}_1$ and $\mathbf{W}_2$ are learnable weight matrices to capture the intricate relationships between multi sensor inputs, $\mathbf{X}$ is the input MTS data, and $d_k$ is the scaling factor. $\sigma$ is the row-wise softmax function to make a normalized adjacency matrix. The relation of each sensor is calculated by considering the temporal features of every sensor. Therefore, depending on the input, the similarity matrix potentially possesses different spatial correlations among multiple sensors.

With the generated graph, the DG layer applies GCN (Kipf & Welling, 2017) to aggregate messages as follows:

$$\mathbf{H}_{\text{dg}} = \mathbf{A}_{\text{dg}}\mathbf{X}\mathbf{W}_{\text{dg}}, \tag{2}$$

where $\mathbf{H}_{\text{dg}} \in \mathbb{R}^{N \times T}$ is the representation obtained via the domain generalization layer, and $\mathbf{W}_{\text{dg}}$ is the weight of a GNN that does not change the dimension. The DG layer adaptively modifies the feature by considering the sensor correlation of the input, accommodating new source data effectively.

Furthermore, we propose a new loss function which aligns the spatial dependency. The domain generalization loss is computed by minimizing the discrepancy between two embeddings obtained from different phase models. $\mathbf{Z}_{\text{ori}}$ ($= Spatial(\mathbf{X})$) is the embedding that passes the learned spatial layer from the first phase model, and $\mathbf{Z}_{\text{dg}}$ ($= Spatial(\mathbf{H}_{\text{dg}})$) is acquired by processing the additional domain generalization layer before the spatial layer ($Spatial$), as shown in Figure 2. In other words, $\mathbf{Z}_{\text{dg}}$ can be interpreted as the embedding of the spatial-wise transformed input data. Finally, the loss term on the domain generalization layer can be formulated like:

$$\mathcal{L}_{\text{dg}} = \|\mathbf{Z}_{\text{ori}} - \mathbf{Z}_{\text{dg}}\|_F^2, \tag{3}$$

where $\|\mathbf{A}\|_F^2$ is the Frobenius norm of matrix $\mathbf{A}$. The loss function can be interpreted as minimizing the variance between two representations: one processed by the DG layer and one that is not processed. This approach facilitates the optimization of the DG layer to achieve better spatial alignment of unseen distributions, enabling the model to learn invariant features with the difference of spatial relations taken into account.

## 3.4 TWO-VIEW REGULARIZATIONS

This section introduces a two-view regularization technique composed of channel decorrelation loss and Gaussian kernel loss. By applying the loss functions to both spatial and temporal layers, the spatiotemporal model can capture essential information specific to each sensor by reducing redundancy from complicated spatial dependency and preserving short-term temporal signals, which helps the DG layer to effectively learn to align multiple spatial dependencies.

**Channel Decorrelation Loss**    While the spatial layer effectively merges information from multiple related sensors, an excessive focus on spatial relationships can lead to over-amplifying the spatial dependencies to the first phase source data, consequently diminishing the model's robustness to distributional shifts. To refine the model's capability in capturing general spatial relations, we propose the channel decorrelation loss. Accordingly, the channel decorrelation loss acts as a regularizer that emphasizes the information specific to each sensor, and it can be expressed as:

$$\mathcal{L}_{\text{spa}} = \|\mathbf{D} - \mathbf{I}\|_F^2, \tag{4}$$

$$\mathbf{D} = \mathbf{Z}\mathbf{Z}^{\mathsf{T}}, s.t. \mathbf{Z} = \{\mathbf{Z}_{\text{ori}}, \mathbf{Z}_{\text{dg}}\} \tag{5}$$

where $\mathbf{D} \in \mathbb{R}^{N \times N}$ is the feature correlation matrix constructed by the hidden representation $\mathbf{Z}$ from the spatial layer. $\mathbf{I}$ in equation 4 is the identity matrix with the same dimension as $\mathbf{D}$. The loss function, which reduces the gap between $\mathbf{D}$ and $\mathbf{I}$, induces each channel to retain its essential information, alleviating the redundant data and noise from the exaggerated consideration of domain-specific spatial correlations. Depending on the phase in which this loss function is applied, what $\mathbf{Z}$ indicates differs: $\mathbf{Z}_{\text{ori}}$ derived from the first phase input, and $\mathbf{Z}_{\text{dg}}$ with the second phase input.

---

**Algorithm 1** Computing the Gaussian kernel loss, $\mathcal{L}_{temp}$

---

1: **Input**: the output of LSTM layer, $\mathbf{C}$

2: **def** GAUSSIAN($d, \sigma = 1.0$):

3:      **return** $\exp\left(-\frac{d^2}{2\sigma^2}\right)$

4: Initialize $dist\_matrix$ and $kernel\_matrix$ with zeros of size $T \times T$
5: $td\_matrix \leftarrow \mathbf{C}^\intercal\mathbf{C}$
6: $dist\_matrix[i,j] \leftarrow \frac{|i-j|}{T}, \; \forall i,j = 0, 1, \ldots, T-1$
7: $kernel\_matrix[i,j] \leftarrow$ GAUSSIAN($dist\_matrix[i,j]$), $\forall i,j = 0, 1, \ldots, T-1$
8: $\mathcal{L}_{temp} \leftarrow \|td\_matrix - kernel\_matrix\|_F^2$

---

**Gaussian Kernel Loss**   The Gaussian kernel loss serves as an additional regularizer to effectively capture and maintain short-term signals, a widely recognized and intuitive approach. This approach facilitates that temporally close representations remain similar while temporally distant representations become progressively more distinct. The Gaussian kernel loss, $\mathcal{L}_{temp}$, can be derived as described in Algorithm 1. In the algorithm, the $td\_matrix$ represents the time dependency matrix, which is similar to temporal correlation matrix, and the $dist\_matrix$ in line 8 denotes a matrix where each element represents the distance from the diagonal of the kernel matrix. By utilizing $dist\_matrix$ as an argument for the GAUSSIAN function, the $kernel\_matrix$ is formulated, with higher values along the matrix's diagonal compared to areas further from the diagonal. The temporal regularization loss minimizes the discrepancy between the time dependency matrix and the kernel matrix. Since $td\_matrix$ is the spatiotemporal representation, regularizing the temporal dependency positively impacts identifying the intrinsic spatial relations among sensors.

### 3.5 TIME COMPLEXITY ANALYSIS

Compared to the existing MTS spatiotemporal models, ASAM additionally proposes a DG layer and three loss functions. The time complexity of each is discussed below. The time complexity of adaptive GNN for DG-layer is $\mathcal{O}(N^2 T)$, where $N$ is the number of channels and $T$ is the dimension of initial features, ensuring efficiency. Additionally, the two-view regularization loss involves $\mathcal{O}(N^2 T)$ for channel decorrelation and $\mathcal{O}(F^2 T)$ for Gaussian kernel loss, with notation corresponding to the main paper. Finally, the time complexity of ASAM is $\mathcal{O}(N^2)$. We note that $N$, which is the number of sensors, is inherently limited by practical constraints, thereby reducing concerns about time complexity.

### 3.6 TRAINING

The two phases of the training process proceed in such a way that one ends and the other begins. For each phase dataset ($\mathcal{D}_{S_{ph1}}$ and $\mathcal{D}_{S_{ph2}}$), batches are constructed regardless of multi-source domain information shuffling all data belonging to the specific phase.

In the first phase, we train the spatiotemporal model ($f_\theta$). Subsequently, we employ a MLP structure, $clf$, to reshape the final input $\mathbf{C}$ to match the number of class labels. The class probability can be represented as:

$$\hat{y} = clf(\text{flatten}(\mathbf{C})), \tag{6}$$

where flatten is a function that reshapes the embedding into a 1-dimensional vector. Finally, we optimize our model with the cross-entropy loss $\mathcal{L}_{ce}$. With the two-view regularization loss defined earlier, our final loss function for the first phase can be denoted as below:

$$\mathcal{L}_{P1} = \mathcal{L}_{ce} + \lambda_1\mathcal{L}_{temp} + \lambda_2\mathcal{L}_{spa}, \tag{7}$$

where $\lambda_1$ and $\lambda_2$ are hyperparameters controlling the impact of both regularization losses.

In the second phase, the new source data initially passes through the domain generalization layer, followed by the spatiotemporal layer utilized in the first phase ($g_{\pi,\theta}$). The relation between two model, $f_\theta$ and $g_{\pi,\theta}$, can be represented like:

$$g_{\pi,\theta}(\mathbf{X}) = f_\theta(a_\pi(\mathbf{X})), \tag{8}$$

where $a_\pi$ indicates the domain generalization layer. An additional invariance loss term is incorporated in this phase. The loss function for the second phase is given by:

$$\mathcal{L}_{\text{P2}} = \mathcal{L}_{\text{ce}} + \lambda_2 \mathcal{L}_{\text{spa}} + \lambda_3 \mathcal{L}_{\text{dg}}, \tag{9}$$

where $\lambda_3$ is the hyperparameter to adjust the effect of invariance loss. In configuring the model, hyperparameters ($\lambda_1$, $\lambda_2$, and $\lambda_3$) were deliberately constrained within a narrow range. This approach was adopted to reduce the model's sensitivity to hyperparameter values. We note that, in inference, class labels are predicted using the second phase model.

### 3.7 THEORETICAL ANALYSIS ON THE DOMAIN GENERALIZATION LAYER

We yield theoretical analysis of the newly proposed loss functions ($\mathcal{L}_{\text{dg}}$ and $\mathcal{L}_{\text{spa}}$). The loss functions can be associated with the information bottleneck principle (Tishby & Zaslavsky, 2015; Wu et al., 2020). Before going into the details, we first introduce additional notations.

**Notations**  $DG$ is the domain generalization adopted signal. $P(X)$ and $P(X|Y)$ denote the distribution of random variable $X$ and the distribution of $X$ conditioned by $Y$. $I(X, Y)$ denotes the mutual information between two random variables $X$ and $Y$. $I(X, Y|A)$ implies the conditional mutual information of $X$ and $Y$ given random variable $A$.

**Theoretical Analysis**  We designate the second phase representation as $\mathbf{Z}_{dg}$. Through the assumption of setting the distribution of $P(\mathbf{Z}_{dg})$ and $P(\mathbf{Z}_{dg}|\mathbf{X})$ as Gaussian, we can lead to the following Theorem:

**Theorem 1.** *Optimizing $\mathcal{L}_{dg}$ and $\mathcal{L}_{spa}$ corresponds to not only maximizing the mutual information between the embeddings from the second phase (domain generalization) and the input $X$ but also minimizing the conditional mutual information of $\mathbf{Z}_{dg}$ and $DG$ conditioned by the input. We can express like:*

$$\min(\mathcal{L}_{dg} + \mathcal{L}_{spa}) \Rightarrow \max I(\mathbf{Z}_{dg}, \mathbf{X}) \text{ and } \min I(\mathbf{Z}_{dg}, DG|\mathbf{X}). \tag{10}$$

The first term on the right-hand side means that the embedding $\mathbf{Z}_{dg}$ expresses the input data $\mathbf{X}$. The second term indicates that $\mathbf{Z}_{dg}$ should contain more invariant information relative to the input $\mathbf{X}$. Satisfying the two terms indicates robustness to domain generalization because the domain generalization layer can transform the data distribution of the second phase into the data distribution of the first phase. The proof of Theorem 1 is in the Appendix. We further define the information bottleneck under the MTS sensor data that suffers from the domain shift.

**Definition 1** (MTS sensor-based information bottleneck)**.** *The multivariate time series (MTS) sensor-based information bottleneck tries to maximize the mutual information between the input and the second phase embedding while minimizing the overlapping information between the domain generalization signal and the corresponding signal. It can be formulated as:*

$$IB_{MTS} = I(X, \mathbf{Z}_{dg}) - \beta I(DG, \mathbf{Z}_{dg}), \tag{11}$$

where $\beta$ is the hyperparameter that controls the mutual information loss. $IB_{\text{MTS}}$ indicates the multivariate time series (MTS) sensor-based information bottleneck. According to Definition 1, satisfying the information bottleneck for sensors is observing the property that is robust to distribution shift.

**Theorem 2.** *If $\beta$ is bounded in [0, 1], then lowering the losses defined on the domain generalization layer is identical to maximizing the sensor information bottleneck.*

$$\min(\mathcal{L}_{dg} + \mathcal{L}_{spa}) \Rightarrow \max IB_{MTS}. \tag{12}$$

Therefore, applying the losses on the domain generalization layer extracts relevant and meaningful information from different distribution data, which in turn enhances the performance of our model. In other words, the variance between two embeddings utilized in the loss function is minimized regardless of the input domain.

Table 1: Overall performance comparison on the MTS sensor dataset, evaluated by accuracy. **Bold-faced** indicates the best result, and underlined results represent the second-best models.

| Dataset | Feature learning | | | | | | Domain generalization | | | | ASAM |
|---|---|---|---|---|---|---|---|---|---|---|---|
| | PICCA | TCN | 2SRNN | SimpleAtt | STCN-GR | GNN-SD | GSAT | CAL | GREA | DisC | (Our model) |
| Ninapro DB 5 | 80.09 | 76.76 | 80.52 | 80.26 | 75.12 | 84.76 | **88.95** | 88.54 | 61.64 | 88.63 | 87.95 |
| SD-gesture | 84.78 | 72.79 | 88.44 | 87.54 | 69.04 | 63.36 | 85.15 | 86.78 | 84.15 | 86.19 | **93.04** |
| HHAR | 65.38 | 64.07 | 60.16 | 71.64 | 72.42 | 70.85 | 52.71 | 69.02 | 67.94 | 67.54 | **74.63** |
| UCI-HAR | 77.77 | 75.53 | 79.81 | 89.87 | 86.37 | 87.59 | 89.00 | 85.69 | 73.74 | 84.11 | **92.42** |
| Average | 77.01 | 72.29 | 77.23 | 82.33 | 75.74 | 76.64 | 78.95 | 82.51 | 71.87 | 81.62 | **86.94** |

Table 2: DG model performance comparison for cross-subject task evaluated by accuracy. The performances of 5 scenarios for each dataset are presented. Columns denote the different target domains for the scenarios. Avg. is the average performance of 5 scenarios. In each column, the **boldfaced** score denotes the best result and underlined results are the second-best models.

| Models | HHAR | | | | | | UCI-HAR | | | | | |
|---|---|---|---|---|---|---|---|---|---|---|---|---|
| | subject 1 | subject 2 | subject 3 | subject 4 | subject 5 | Avg. | subject 1 | subject 2 | subject 3 | subject 4 | subject 5 | Avg. |
| GSAT | 35.98 | 48.85 | **63.56** | 62.85 | 52.33 | 52.71 | 94.70 | **96.48** | **88.64** | 80.73 | 84.46 | 89.00 |
| CAL | **59.16** | 82.29 | 36.06 | 80.67 | 86.94 | 69.02 | 89.40 | 76.54 | 82.65 | 89.73 | 90.15 | 85.69 |
| GREA | 50.68 | 83.69 | 60.78 | 63.18 | 81.37 | 67.94 | 55.96 | 54.25 | 86.12 | 81.64 | 90.77 | 73.74 |
| DisC | 40.18 | 90.25 | 39.15 | 82.87 | 85.26 | 67.54 | 94.37 | 84.75 | 84.54 | 82.45 | 74.46 | 84.11 |
| ASAM | 52.03 | **92.44** | 53.26 | **83.7** | **91.74** | **74.63** | **97.36** | 92.96 | 83.91 | **92.21** | 95.69 | **92.42** |

# 4 EXPERIMENT

## 4.1 DATASET AND BASELINES

We experiment with four datasets: two surface electromyography (sEMG) sensor datasets for hand gesture classification and two human activity recognition (HAR) datasets for human activity recognition. Ninapro DB5, which includes 6 repetitions and 53 hand gesture labels, and SD-gesture, which includes 4 repetitions and 18 static and dynamic hand gesture labels, which are proposed in (Lee et al., 2023). Furthermore, we utilized two additional human activity recognition datasets (HHAR and UCI-HAR) to compare the accuracy of the human activity recognition task.

We implemented 10 recent baselines, which are categorized into two parts: baselines related to feature learning (Canonical correlation analysis-based model (Cheng et al., 2018) (PICCA), TCN-based sEMG model (Tsinganos et al., 2019), Simple Attention (Josephs et al., 2020) (SimpleAtt for short), RNN-based model with simple multi-step domain adaptation (Ketykó et al., 2019) (2SRNN), GNN-based model STCN-GR (Lai et al., 2021), self-attention graph generation model for static and dynamic gestures (Lee et al., 2023) (GNN-SD for short)), and the other for the domain generalization methods especially in the graph domain (GSAT (Miao et al., 2022), CAL (Sui et al., 2022), GREA (Liu et al., 2022), and DisC (Fan et al., 2022)). The table, which denotes the comparison between recent domain generalization methods and ASAM and details about the datasets and baselines, are further discussed in the Appendix.

## 4.2 EXPERIMENTAL SETTING

We evaluated ASAM on various experiment settings, including the cross-repetition and cross-subject. The cross-repetition evaluation is applied to the sEMG data since the data distribution varies even for the same patient. Specifically, the cross-repetition task trains the model with the source repetition data from the same subject and aims to have robust performance on unobserved target repetitions from the same subject. The human activity recognition dataset is evaluated with the cross-subject setting, where each domain is defined as a subject. One subject is defined as the target domain, and the remaining subjects are treated as the source domain. The remaining source domain is split into half for phase 1 and phase 2 domains, respectively. We sampled 5 subjects as the target subjects in both the HHAR and UCI-HAR datasets, and the averaged accuracy of the performance of 5 subjects is compared with the baselines. Further details about the dataset is discussed in the Appendix.

Table 3: Ablation study of accuracy comparison for the loss function, domain generalization layers, temporal layers, and spatial layers on ASAM.

| (a) Loss function | SD-gesture | UCI-HAR |
|---|---|---|
| w/o $\mathcal{L}_{spa} + \mathcal{L}_{temp}$ | 90.78 | 80.65 |
| w/o $\mathcal{L}_{spa}$ | 91.99 | 89.98 |
| w/o $\mathcal{L}_{temp}$ | 92.40 | 87.47 |
| ASAM | 93.04 | 92.42 |

| (b) DG layer | SD-gesture | UCI-HAR |
|---|---|---|
| w/o DG layer | 91.00 | 85.20 |
| Linear layer | 85.41 | 80.88 |
| MLP layer | 83.67 | 88.26 |
| ASAM | 93.04 | 92.42 |

| (c) Temporal layer | SD-gesture | UCI-HAR |
|---|---|---|
| TCN | 65.88 | 79.75 |
| Transformer | 83.28 | 85.38 |
| ASAM | 93.04 | 92.42 |

| (d) Spatial layer | SD-gesture | UCI-HAR |
|---|---|---|
| CNN | 88.13 | 82.45 |
| Attention | 83.29 | 86.09 |
| ASAM | 93.04 | 92.42 |

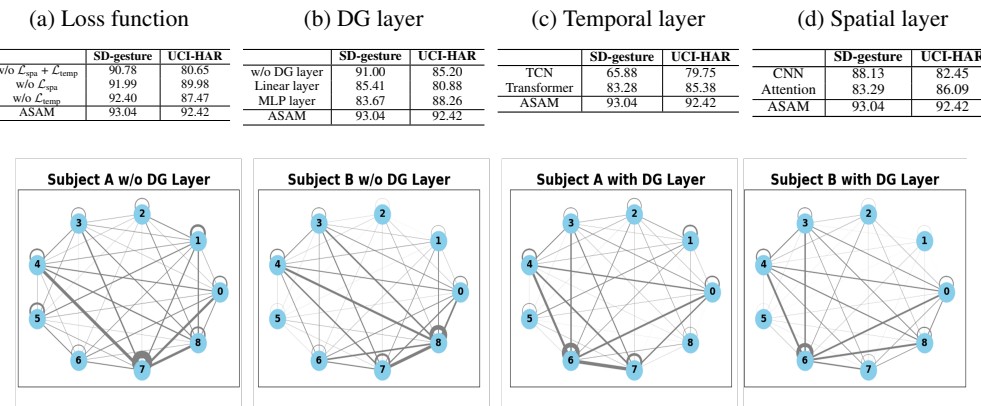

Figure 3: Graphs generated by incorporating the DG layer for two domains. The edge indicates the spatial dependency between the two sensors, and its thickness denotes the magnitude. Subjects A and B are randomly selected from $\mathcal{D}_{S_{ph2}}$ of the UCI-HAR dataset.

## 4.3 PERFORMANCE COMPARISON

This section presents the results of our proposed model. Our model demonstrates strong performance across four datasets, as shown in Table 1, underscoring its effectiveness of spatial alignment in MTS sensor-related classification tasks. The table indicates that the effects of previous feature learning and domain generalization methods vary depending on the dataset. However, our ASAM method significantly alleviates the distribution shift issue in the spatiotemporal domain across various experimental settings, including both the cross-repetition and cross-subject scenarios. ASAM outperforms the best competitor by 4.43% on the average performance of all datasets. For the Ninapro DB5 dataset, a different domain generalization method slightly outperforms ASAM, but the performance difference is marginal.

Furthermore, Table 2 exhibits the detailed experiment of the HAR dataset compared with other domain generalization baselines. The existing domain generalization baselines are not robust on the HAR dataset, leading to underperformance even compared to the feature learning baselines. As shown in the table, ASAM outperforms most baselines in various target domain scenarios (subjects). Specifically, for the UCI-HAR dataset, our model shows consistent results on various scenarios compared to others, indicating robustness. The performance table with the standard deviation is included in the appendix.

## 4.4 ABLATION STUDY

In this section, we conduct an ablation study to verify ASAM's effectiveness. We illustrate the generated graph to show the impact of the DG layer that aligns the spatial dependency of domains. We examine the impact of additional losses, the GNN-based domain generalization model, and different temporal layers. We also perform a sensitivity analysis on hyperparameters.

**Visualization of the Generated Graph**   We visualize the generated graph on the DG layer to confirm its effectiveness in capturing spatial alignment against distribution shifts. The edges between sensors are established by calculating the cosine similarity between their representations. In other words, each edge in the visualized graph represents the spatial dependency between two sensors, with the edge width indicating the strength of dependency. We utilized the graphs before and after processing through the DG layer, as shown in Figure 3. The figure shows that the input graph (without the DG layer) varies across distinct subjects (domains) but remains consistent after processing through the domain generalization layer. These results indicate that our domain generalization effectively aligns the spatial dependencies across different data distributions.

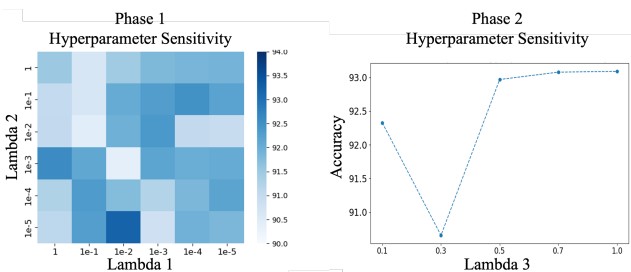

Figure 4: Hyperparameter sensitivity of the loss functions on the SD-gesture dataset. Specifically, we illustrated $\lambda_1, \lambda_2$ on the spatiotemporal model and the effect of $\lambda_3$ on the domain generalization model.

**Analysis of Loss Functions**    Table 3 demonstrates the effectiveness of the newly proposed loss functions. The table shows that the spatial layer's regularization loss ($\mathcal{L}_{\text{spa}}$) and the temporal layer's regularization loss ($\mathcal{L}_{\text{temp}}$) positively impact performance. This result indicates that employing additional loss functions is beneficial for capturing spatial alignment across domains more effectively.

**Discussions of GNN-based DG Layer**    An ablation study was designed to compare the performance of various domain generalization layer variants. Table 3 compares our model without the domain generalization layer, our model with a single MLP layer for DG, and our model with three MLP layers for DG. The results show that ASAM consistently outperforms the other settings. In particular, when the number of sensors is small, as in the SD-gesture and UCI-HAR datasets, ASAM, which utilizes the input-driven domain generalization layer, significantly improves performance. An interesting observation is that employing an adequate domain generalization layer is essential. Applying linear or multiple linear structures even underperforms compared to the absence of a domain generalization layer. Therefore, capturing spatial relations with the GNN-based domain generalization layer enhances the capability to align spatial dependencies that are robust to the distribution shift.

**Effects of Variations on Temporal and Spatial Layer**    ASAM adopts the LSTM structure for the temporal layer. However, there are more advanced sequential models, such as Transformer (Vaswani et al., 2017) and TCN (Bai et al., 2018). Table 3 supports the effectiveness of applying LSTM to our temporal data. We conducted a comparison by substituting the temporal layers with Transformer or TCN structures. The performance of the Transformer and TCN models is comparatively lower than that of the LSTM, as shown in Table 3. Similarly, we explore variations of the spatial layer by adopting CNN- and Attention-based models. As shown in the table, the simple GNN-LSTM structure consistently outperforms these variations, achieving the best performance.

**Analysis of Hyperparameter Sensitivity**    We conducted an experiment to establish the effect of different loss functions by adjusting the $\lambda$ values introduced in Section 3.6. Two experiments were designed: sensitivity analysis on the phase 1 model (spatiotemporal model) and the phase 2 model (domain generalization model). The left figure of Figure 4 shows a heatmap of different $\lambda$ values. The results show variations in accuracy, with the best performance achieved when $\lambda_1$ is set to 0.01 and $\lambda_2$ to 0.00001. We also experimented with the effect of $\lambda_3$ by fixing $\lambda_2$ at its optimal value. The accuracy differences across various hyperparameters of $\lambda_3$ are shown on the right side of Figure 4.

## 5    CONCLUSION

We proposed a domain generalization method for MTS data with spatial consideration to mitigate the distribution shift issue. Additionally, we introduced new loss functions aimed at aligning spatial dependencies with domains. We also provided theoretical insights indicating that our domain generalization layer and its corresponding loss functions optimally induce the MTS-induced information bottleneck. The proposed model consistently outperforms most baseline models across four datasets. To our knowledge, this work is one of the first to investigate domain generalization approaches for MTS data, specifically emphasizing the alignment of spatial dependencies.

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

Table 4: Comparison to other domain generalization models.

|  | GSAT | CAL | GREA | DisC | ASAM |
|---|---|---|---|---|---|
| **Extracting graph structure** | ○ | ○ | ○ | ○ | ○ |
| **Using regularization constraint** | ○ | × | ○ | × | ○ |
| **Learning attention weight** | Edge-level | Edge-level | Node-level | Edge-level | Graph-level |
| **Learning w/o input graph** | × | × | × | × | ○ |
| **Adaptively aligning spatial dependencies** | × | × | × | × | ○ |

# A   APPENDIX

## A.1   EXPERIMENTAL SETUP

### A.1.1   DATASETS

**Ninapro** is a widely-used sEMG dataset for hand gesture recognition task (Josephs et al., 2020; Ketykó et al., 2019; Côté-Allard et al., 2017). It comprises 10 large databases; we used Ninapro DB 5, which includes data from two MYO armband sensors (16 channels total). The dataset is divided into Exercises A, B, and C. We utilized the full DB 5 (53 gestures) from 10 subjects, with each gesture repeated 6 times.

**SD-gesture** is a sparse sEMG dataset with only 8 channels (Lee et al., 2023). The paper that proposed this dataset focuses on static and dynamic gestures, and our paper names this dataset SD-gestures. The dynamic gesture indicates that the sensor data of its gestures are acquired when the subject performs a specific action repeatedly, while the conventional static gesture is acquired by maintaining the gestures. It is comprised of 18 gestures (14 static, 4 dynamic). This dataset has 4 repetition data on 9 different subjects.

**HHAR** is a human activity recognition dataset which comprises data from 9 subjects, gathered using 3-channel structured sensors from smartphones and smartwatches (Stisen et al., 2015). It contains 6 human activities as classes.

**UCI-HAR** is composed of data collected using three different type of sensors: an accelerometer, a gyroscope, and body sensors, which were utilized on 30 subjects (Anguita et al., 2013). Each subject executed six distinct activities, including walking, walking upstairs, walking downstairs, standing, sitting, and lying down.

### A.1.2   BASELINE MODELS

We compared ASAM with 10 recent baselines. Each model is described in the following.

PICCA (Cheng et al., 2018) is a model that utilizes Canonical Correlation Analysis (CCA) for the sEMG domain, maximizing the correlation between the training repetition data and the domain generalization repetition data. TCN (Tsinganos et al., 2019) applies a TCN structure, a widely used architecture for sequential data. 2SRNN (Ketykó et al., 2019) is an RNN-based model that utilizes domain generalization. This model applies a single linear layer to adapt the domain generalization repetition into the training repetition. It can learn universal representations by applying a two-phase design similar to ASAM. SimpAtt (Josephs et al., 2020) harnesses an attention mechanism to identify temporal relations. It first expands the number of sensor dimensions to a higher dimension to enrich the representations. Then, it applies the attention mechanism to capture temporal relations for identifying hand gestures. STCN-GR (Lai et al., 2021) is a GNN-based model. Each edge is treated as a learnable parameter with a range of [0, 1]. The model is divided into a spatial layer and a temporal layer, which are GNN-based and TCN-based, respectively. GNN-SD (Lee et al., 2023) is the model that creates a new adjacency matrix based on the input value. This model also splits the spatial layer and temporal layer. For the spatial layer, they similarly utilize the graph generation phase and the GNN phase. For the temporal layer, they applied a CNN-based graph with a GLU layer that reduces the redundancy of the feature matrix.

Additionally, graph domain generalization methods (GSAT (Miao et al., 2022), CAL (Sui et al., 2022), GREA (Liu et al., 2022), and DisC (Fan et al., 2022)) are adopted as the baselines. GSAT harnesses the information bottleneck principle to detect task-relevant subgraphs. CAL discovers the causal pattern and mitigates the graph's confounding effect. GREA utilizes data augmentation to

Table 5: Hyperparameters to reproduce our results.

| Hyperparamter | Ninapro DB 5 | SD-gesture | HHAR | UCI-HAR |
|---|---|---|---|---|
| lr | 0.001 | 0.001 | 0.001 | 0.001 |
| batch size | 512 | 1024 | 512 | 512 |
| $\lambda_1$ | 1e-2 | 1e-1 | 1e-3 | 1e-3 |
| $\lambda_2$ | 1e-5 | 1e-4 | 1e-3 | 1e-3 |
| $\lambda_3$ | 0.5 | 1.0 | 1.0 | 0.5 |
| $\sigma$ | 3.0 | 5.0 | 1.0 | 1.0 |
| $F$ | 512 | 512 | 512 | 512 |

improve task-relevant information. DisC disentangles the causal and bias graphs by extracting the spurious correlations in the graph. The comparison between recent domain generalization methods and ASAM is shown in Table 4. ASAM utilizes input-driven graph generation to analyze the entire graph during critical subgraph extraction, while most baselines assign weights to edges independently to isolate task-relevant subgraphs. Additionally, ASAM uniquely learns sensory inputs without requiring a predefined physical relationship graph. Finally, ASAM is the only model that can adaptively align the spatial dependencies to unseen domains.

### A.1.3 HYPERPARAMETERS

To ensure the reproducibility of our model, we list the hyperparameters used in ASAM in Table 5. These parameters were selected based on the validation splits across all datasets. The search space for the learning rate included [0.0001, 0.001, 0.01, 0.1], and for batch size, we considered [128, 256, 512, 1024]. For hyperparameters related to the loss functions ($\lambda_1$, $\lambda_2$, and $\lambda_3$), our search ranged from 1e-6 to 1, increasing by a factor of ten at each step. For $\sigma$, we explored values from 1.0 to 10.0 in increments of 1. The embedding dimension of the LSTM ($F$) experimented with values ranging from 32 to 1024, specifically using powers of 2 within this range.

### A.1.4 SPATIO-TEMPORAL MODEL

**Spatial Layer** We propose a GNN-based model that can capture every sensor's spatial relations. The graph is generated by regarding sensor channels as nodes and the relations between two different sensors as edges. Inspired by the work in (Lee et al., 2023), the graph is constructed with a feature matrix. Consequently, the edge weight between sensors with similar features is designed to be higher, while it is lower for those with dissimilar features. The adjacency matrix for the spatial layer $A_s \in \mathbb{R}^{N \times N}$ can be expressed as follows:

$$A_s = \sigma(\frac{(\mathbf{W}_Q \mathbf{X})(\mathbf{W}_K \mathbf{X})^\mathsf{T}}{\sqrt{d_k}}), \tag{13}$$

where $\mathbf{W}_Q$ and $\mathbf{W}_K$ are learnable weight matrices to capture the intricate relationships between multi sensor inputs, $\mathbf{X}$ is the input MTS data, and $d_k$ is the scaling factor. $\sigma$ is the row-wise softmax function to make a normalized adjacency matrix.

We adopt the Graph Neural Network (GNN) model (Kipf & Welling, 2017) for capturing spatial dependencies. The primary distinction between the GNN model and ASAM lies in our use of a soft adjacency matrix, implying that our matrix behaves as if it has varying edge weights. The aggregated features from the generated graph can be represented as:

$$\mathbf{h}^l = A_s \mathbf{W}_l \mathbf{h}^{l-1} + \mathbf{h}^{l-1}, \tag{14}$$

where $\mathbf{h}^l \in \mathbb{R}^{N \times T}$ is the hidden representation on the $l$-th layer, $\mathbf{W}_l$ as the weight parameter that does not change the dimension. $\mathbf{h}^0$ is identical to the sensor input $\mathbf{x}$, and $\mathbf{h}^L$ is $\mathbf{Z}$. We set the number of GNN layer $L$ as 2 for the experiment.

**Temporal Layer** In the temporal layer of our model, we utilize the Long Short-Term Memory (LSTM) layer (Hochreiter & Schmidhuber, 1997) to have the temporal representations. It's worth noting that while recent advanced temporal models like the Transformer (Vaswani et al., 2017) and TCN (Bai et al., 2018) can be applied, they tend to overfit to the main repetition, leading to a

Table 6: Comparison of execution time per epoch of baselines and ASAM.

| Model | ASAM | PICCA | TCN | 2SRNN | SimpleAtt | STCN-GR | GNN-SD | GSAT | CAL | GREA | DisC |
|-------|------|-------|-----|-------|-----------|---------|--------|------|-----|------|------|
| HHAR | 2.94 | 1.54 | 3.52 | 1.21 | 0.32 | 3.44 | 3.55 | 2.88 | 4.04 | 3.97 | 4.01 |
| UCI-HAR | 1.69 | 1.07 | 2.86 | 0.32 | 0.28 | 2.59 | 2.77 | 0.67 | 0.89 | 0.91 | 0.89 |

Table 7: Overall performance comparison on the MTS sensor dataset with the standard deviation. **Boldfaced** indicates the best result, and underlined results represent the second-best models.

| Model | Ninapro DB 5 | SD-gesture | HHAR | UCI-HAR | Average |
|-------|--------------|------------|------|---------|---------|
| PICCA | 80.09 ± 0.24 | 84.78 ± 0.78 | 65.38 ± 23.93 | 77.77 ± 13.39 | 77.01 |
| TCN | 76.76 ± 2.41 | 72.79 ± 0.23 | 64.07 ± 12.80 | 75.53 ± 11.66 | 72.29 |
| 2SRNN | 80.52 ± 0.21 | 88.44 ± 0.55 | 60.16 ± 29.01 | 79.81 ± 21.6 | 77.23 |
| SimpleAtt | 80.26 ± 0.37 | 87.54 ± 0.69 | 71.64 ± 14.75 | 89.87 ± 9.37 | 82.33 |
| STCN-GR | 75.12 ± 0.16 | 69.04 ± 2.54 | 72.42 ± 13.22 | 86.37 ±6.60 | 75.74 |
| GNN-SD | 84.76 ± 0.29 | 63.36 ± 0.59 | 70.85 ± 17.35 | 87.59 ± 5.46 | 76.64 |
| GSAT | **88.95 ± 0.22** | 85.15 ± 0.81 | 52.71 ± 10.15 | 89.00 ± 5.96 | 78.95 |
| CAL | 88.54 ± 0.09 | 86.78 ± 0.86 | 69.02 ± 19.06 | 85.69 ± 5.35 | 82.51 |
| GREA | 61.64 ± 0.09 | 84.15 ± 4.01 | 67.94 ± 12.65 | 73.74 ± 15.50 | 71.87 |
| DisC | 88.63 ± 0.23 | 86.19 ±0.41 | 67.54 ± 22.89 | 84.11 ± 6.35 | 81.62 |
| **ASAM** | 87.95 ± 0.43 | **93.04 ± 1.03** | **74.63 ± 18.22** | **92.42 ± 4.64** | **86.94** |

degraded performance on the test dataset due to distribution differences. The temporal layer can be expressed as shown below:

$$\mathbf{C} = \text{LSTM}(\mathbf{Z}), \tag{15}$$

where $\mathbf{C} \in \mathbb{R}^{F \times T}$ is the output of the LSTM layer, where $F$ is the output dimension of the LSTM layer.

### A.1.5 EXTRA EXPERIMENTAL SETTING

Every experiment was conducted on an NVIDIA-RTX A6000 GPU. The average runtime of each dataset is compared using the average runtime per epoch for a fair comparison. The total number of trained epochs differs for each dataset due to the early stopping technique we applied. We compared the accuracy by setting the seed to 0 to 5 for consistency. The average runtime per epoch for the Ninapro DB5 and SD-gesture datasets is 6.15 seconds and 2.77 seconds, respectively. Similarly, the HHAR and UCI-HAR datasets took 2.94 seconds and 1.69 seconds per epoch. The runtime depends on the dataset size, with the largest dataset, Ninapro DB5, requiring the most time. The per-epoch execution time comparison is presented in Table 6. While the execution time of ASAM is not the shortest among the methods evaluated, it remains comparable to the baselines and demonstrates efficiency within a practical range. This indicates that ASAM achieves competitive runtime performance without incurring excessive computational overhead.

### A.1.6 PERFORMANCE WITH THE STANDARD DEVIATION

We provide the performance table derived from Table 1 that also includes the standard deviation and is expressed in Table 7. From the table, we can observe that in the dataset like UCI-HAR, our model is more robust indicating that aligning the multiple spatial dependencies is crucial in MTS sensor data.

### A.1.7 EXTENDED EXPERIMENT ON HHAR DATASET

We further discuss the limitations of the feature-learning-based methods in the HHAR dataset. For each evaluation subject (i.e., domain), we compute the average JS-divergence or average accuracy over all other subjects. A high score indicates a larger distribution discrepancy. Table 8 implies that when the distribution gap is high between the training data and evaluation data, which is subject 1 or subject 3 in the table, feature-based methods underperform compared to ours.

Table 8: Performance comparison with the feature learning baselines on the HHAR dataset with Jensen-Shannon divergence score.

| | Subject 1 | Subject 2 | Subject 3 | Subject 4 | Subject 5 |
|---|---|---|---|---|---|
| JS-divergence | 0.1304 | 0.0868 | 0.1024 | 0.0978 | 0.0981 |
| 2SRNN | 24.00 | 81.15 | 27.77 | 91.32 | 72.64 |
| SimpleAtt | 39.40 | 88.80 | 52.63 | 89.55 | 87.84 |
| STCN-GR | 42.64 | 93.00 | 45.91 | 92.37 | 72.42 |
| ASAM | 52.03 | 92.44 | 53.26 | 83.70 | 91.74 |

Table 9: Performance comparison on the MTS sensor anomaly detection dataset SMAP with recall value.

| | 2SRNN | SimpleAtt | STCN-GR | GSAT | CAL | DisC | ASAM |
|---|---|---|---|---|---|---|---|
| SMAP | 67.76 | 70.68 | 67.53 | 0 | 0 | 0 | **70.79** |

### A.1.8 ANOMALY DETECTION EXPERIMENT

To strengthen the difference between existing DG methods and ours, we conduct additional experiment on MTS-based anomaly detection in the context of multi-domain. SMAP (Hundman et al., 2018) is an expert-labeled telemetry anomaly data. From the data, we defined the domain that has the same channel ID. The dataset is comprised of 25 sensors, and the goal is to detect anomaly sensor signals. We compare the recall values, which indicate how effectively the models detect anomalies, against six baselines that achieved at least the second-highest performance in our main experiment (Table 1). Table 9 exhibits the performance of anomaly detection in terms of a recall value on six baselines. The results presented in the table demonstrate that while domain generalization-based models fail to detect abnormal classes effectively, ASAM performs reasonably in identifying anomalies.

### A.1.9 EXTENDED EXPERIMENT ON BOILER DATASET

To further demonstrate the versatility of ASAM, we extended our experiments to the Boiler (Shohet et al., 2019) dataset, as the datasets used in our main experiments predominantly fall under human activity recognition tasks. The results, presented in Table 10, highlight the potential of ASAM in non-human-related domains.

The Boiler dataset is commonly utilized in unsupervised domain adaptation tasks, where the objective is to adapt a single source domain to a single target domain, often yielding near-perfect accuracy. However, since ASAM operates within a domain generalization (DG) framework, which requires multiple source domains for generalization, we structured the experiment as follows: the model was trained using two domains and evaluated on a separate, unseen domain. This approach inherently introduces more diverse domain information during training. Notably, the accuracy of all methods, including ASAM, approached 100, suggesting that the Boiler dataset poses relatively low complexity in a DG setting. Despite this, the results reinforce that ASAM is adaptable across diverse domains and can be effectively applied to settings beyond human activity recognition.

### A.2 EXTENDED THEORETICAL ANALYSIS AND PROOFS

We demonstrate the proofs of Theorem 1 and Theorem 2 in the theoretical analysis section 3.7 of the main text. We denote $\mathbf{Z_X}$ as the first phase representation. The relationship between the domain generalization loss and the entropy of domain generalized representation ($H(\mathbf{Z}_{\mathrm{dg}}|\mathbf{X})$) is assumed as:

$$\min \mathcal{L}_{\mathrm{dg}} \approx \min H(\mathbf{Z}_{\mathrm{dg}}|\mathbf{X}). \tag{16}$$

equation 16 is to minimize $\mathcal{L}_{\mathrm{dg}}$ and is similar to minimizing the entropy of representation adopted from DG conditioned by the input. $H(\mathbf{Z_X}|\mathbf{X}) = 0$) holds since the representation is deterministic if $\mathbf{X}$ is given. Therefore, diminishing the invariance loss has the effect of impelling similar expressions

Table 10: Performance comparison on the MTS sensor data besides HAR and hand gesture recognition dataset.

|  | 2SRNN | SimpleAtt | STCN-GR | GSAT | CAL | DisC | ASAM |
|---|---|---|---|---|---|---|---|
| Boiler | 99.67 | 99.09 | 99.73 | **99.93** | 92.23 | 92.16 | 99.87 |

between the representation with the original input ($\mathbf{Z_X}$) and the domain generalization representation ($\mathbf{Z}_{\mathrm{dg}}$), is minimizing the right side of equation 16.

Moreover, an extra equation related to the proof is represented as:

$$\min \mathcal{L}_{\mathrm{spa}} \approx \max H(\mathbf{Z}_{\mathrm{dg}}), \tag{17}$$

indicating that the spatial decorrelation loss is maximizing the entropy of domain-generalized representation $\mathbf{Z}_{\mathrm{dg}}$. If we assume that $\mathbf{Z}_{\mathrm{dg}}$ obeys a gaussian distribution, the entropy of $\mathbf{Z}_{\mathrm{dg}}$ can be transformed as:

$$
\begin{aligned}
H(\mathbf{Z}_{\mathrm{dg}}) &= -\int p(\mathbf{Z}_{\mathrm{dg}}) \log p(\mathbf{Z}_{\mathrm{dg}}) d\mathbf{Z}_{\mathrm{dg}} \\
&= -\mathbb{E}[\log \mathcal{N}(\mu_{\mathrm{dg}}, \mathbf{\Sigma}_{\mathrm{dg}})] \\
&= -\mathbb{E}[\log[(2\pi)^{-D/2}|\mathbf{\Sigma}_{\mathrm{dg}}|^{-1/2}\exp(-\frac{1}{2}(\mathbf{Z}_{\mathrm{dg}} - \mu_{\mathrm{dg}})^{\intercal} \\
&\quad \mathbf{\Sigma}_{\mathrm{dg}}^{-1}(\mathbf{Z}_{\mathrm{dg}} - \mu_{\mathrm{dg}}))]] \\
&= \frac{D}{2}\log(2\pi) + \frac{1}{2}\log|\mathbf{\Sigma}_{\mathrm{dg}}| + \frac{1}{2}\mathbb{E}[(\mathbf{Z}_{\mathrm{dg}} - \mu_{\mathrm{dg}})^{\intercal} \\
&\quad \mathbf{\Sigma}_{\mathrm{dg}}^{-1}(\mathbf{Z}_{\mathrm{dg}} - \mu_{\mathrm{dg}})] \\
&= \frac{D}{2}(1 + \log(2\pi)) + \frac{1}{2}|\mathbf{\Sigma}_{\mathrm{dg}}|,
\end{aligned}
\tag{18}
$$

where $\mu_{\mathrm{dg}}, \mathbf{\Sigma}_{\mathrm{dg}}$ denotes the mean and the covariance of $\mathbf{Z}_{\mathrm{dg}}$. $|\mathbf{\Sigma}_{\mathrm{dg}}|$ is the determinant of the covariance matrix of $\mathbf{Z}_{\mathrm{dg}}$. Therefore, maximizing the entropy is identical to maximizing the covariance matrix. If we assume that $\lambda_1, \lambda_2, ..., \lambda_N$ are the $N$ eigenvalues of $\mathbf{\Sigma}_{\mathrm{dg}}$, then $\sum_{i=1}^{N} \lambda_i = \mathrm{trace}(\mathbf{\Sigma}_{\mathrm{dg}}) = N$. Finally, we have the following equation:

$$\log|\mathbf{\Sigma}_{\mathrm{dg}}| = \log \Pi_{i=1}^{N}\lambda_i = \sum_{i=1}^{N}\log \lambda_i \leq N \log \frac{\sum_{i=1}^{N}\lambda_i}{N} = 0. \tag{19}$$

The inequality is due to the Jensen's Inequality. The equation indicates that the upper bound of $|\mathbf{\Sigma}_{\mathrm{dg}}|$ is 1, and it is satisfied when the covariance matrix is the identity matrix. Therefore, maximizing the entropy of domain-generalized representation matches with optimizing the spatial decorrelation loss.

Restate Theorem 1:

**Theorem 1.** *Optimizing $\mathcal{L}_{dg}$ and $\mathcal{L}_{spa}$ corresponds to not only maximizing the mutual information between the embeddings from phase 2 (domain generalization) and the input $X$ but also minimizing the conditional mutual information of $\mathbf{Z}_{dg}$ and $DG$ conditioned by the input.*

*Proof.* From equation 16 and two facts: $I(\mathbf{Z}_{\mathrm{dg}}, \mathrm{DG}|\mathbf{X}) = H(\mathbf{Z}_{\mathrm{dg}}|\mathbf{X}) - H(\mathbf{Z}_{\mathrm{dg}}|\mathrm{DG}, \mathbf{X})$, and $H(\mathbf{Z}_{\mathrm{dg}}|\mathrm{DG}, \mathbf{X}) = 0$ since it is deterministic, we can conclude that our invariance loss is minimizing the mutual information $I(\mathbf{Z}_{\mathrm{dg}}, \mathrm{DG}|\mathbf{X})$. Moreover, from equation 16, equation 17 and the following equation $I(\mathbf{Z}_{\mathrm{dg}}, \mathbf{X}) = H(\mathbf{Z}_{\mathrm{dg}}) - H(\mathbf{Z}_{\mathrm{dg}}|\mathbf{X})$, we can also state that the newly proposed loss functions for the domain generalization is maximizing the mutual information $I(\mathbf{Z}_{\mathrm{dg}}, \mathbf{X})$. Therefore, we conclude the proof since it satisfies the equation 10. □

Restate Theorem 2:

**Theorem 2.** *If $\beta$ is bounded in [0, 1], then lowering the losses defined on the domain generalization layer is identical to maximizing the MTS information bottleneck.*

*Proof.* For the Theorem 2, we can decompose the MTS-based information bottleneck as follows:

$$IB_{\text{MTS}} = [H(\mathbf{Z}_{\text{dg}}) - H(\mathbf{Z}_{\text{dg}}|\mathbf{X})] - \beta[H(\mathbf{Z}_{\text{dg}}) - H(\mathbf{Z}_{\text{dg}}|\text{DG})]. \tag{20}$$

Since $H(\mathbf{Z}_{\text{dg}}|\text{DG}) = 0$ because the representation is deterministic, we can express like:

$$IB_{\text{MTS}} = (1 - \beta)H(\mathbf{Z}_{\text{dg}}) - H(\mathbf{Z}_{\text{dg}}|\mathbf{X}), \tag{21}$$

where maximizing the information bottleneck is not only maximizing $H(\mathbf{Z}_{\text{dg}})$ but also minimizing $H(\mathbf{Z}_{\text{dg}}|\mathbf{X})$. With the equation 16 and equation 17, we complete the proof. □

