# OpenReview forum: "Harnessing Spatial Dependency for Domain Generalization in Multivariate Time-series Sensor Data"
_ICLR.cc/2025/Conference — Submitted to ICLR 2025_

### Official Review · Reviewer_egig · 2024-10-26

**Soundness:** 2
**Presentation:** 2
**Contribution:** 2
**Rating:** 3
**Confidence:** 5

**Summary:**

This paper proposes dynamic contrastive learning method for time series representation learning, which  harness every time step in a sequence as positive and negative pairs. Experiments are carried on clustering and classification tasks.

**Strengths:**

1. The paper is well-structured and clear.
2. theoretical insights and codes are provided.

**Weaknesses:**

1. The abstract is too long and lack readability.

2. The novelty and contribution of this work is unclear and need more clear presentation

3. Comparison and discussion of some relevant methods are needed, including:

   [1] SEA++: Multi-Graph-based High-Order Sensor Alignment for Multivariate Time-Series Unsupervised Domain Adaptation[J]. arXiv preprint arXiv:2311.10806, 2023.

   [2] CauDiTS: Causal Disentangled Domain Adaptation of Multivariate Time Series[C]//Forty-first International Conference on Machine Learning.

   [3] Source-free domain adaptation with temporal imputation for time series data, ACM SIGKDD Conference on Knowledge Discovery and Data Mining. 2023

4. What is the motication of Channel Decorrelation Loss? Variable-wise correlations commonly exist in MTS, and they are representative properties.

5. More datasets should be evaluated in the experiments, e.g., SleepEDF, Boiler ...

6. The experimental setting is unclear.

7. Model computational effort analysis and experiments should be provided.

**Questions:**

Please refer to weakness

---

> ### Author Response · Authors · 2024-11-19
> **Response to Reviewer egig [1/3]**
>
> We sincerely appreciate the reviewer’s insightful feedback and have made every effort to address the identified concerns and weaknesses in a thorough and thoughtful manner.
>
> ## **Abstract**:
> Thank you for your feedback regarding the abstract’s length and readability. We apologize for any inconvenience this may have caused. In response, we have revised the abstract to enhance its clarity and conciseness. The updated version is more focused and succinct, aiming to provide a clearer summary of our work. We appreciate your constructive comments, which have helped us improve the manuscript.
>
> ## **Novelty and contribution**:
> Thank you for your feedback regarding the clarity of our work’s novelty and contributions. We address a significant challenge in domain generalization (DG) for multivariate time-series (MTS) sensor data: existing methods inadequately handle domain-specific spatial dependencies. They often assume a fixed, invariant spatial structure and overlook distributional discrepancies caused by variations in sensor relationships across domains (e.g., due to sensor misalignment or reattachment), as illustrated in Figure 1.
>
> To overcome these limitations, we propose:
>
> ### 1.	Adaptive Spatial Dependency Alignment:
> - A novel Domain Generalization (DG) layer that adaptively models and aligns spatial correlations across different domains without assuming a unified spatial structure.
>
> ### 2.	New Loss Functions Grounded in Information Bottleneck Theory:
> - A domain generalization loss to align spatial dependencies between domains.
> - Two-view regularization losses (spatial decorrelation loss and Gaussian kernel loss) to enhance the capture of underlying spatiotemporal information and prevent overfitting.
> - We demonstrate that our DG layer and loss functions correspond to maximizing the information bottleneck for MTS data, ensuring robustness to domain shifts.
>
> ### 3.	Empirical Validation:
> - Extensive experiments on four real-world datasets (including sEMG and human activity recognition) show that our method outperforms ten recent baselines.
> - Visualizations confirm that the DG layer effectively aligns distinct distributions, resulting in robust performance on unseen domains.
> Our work is among the first to adaptively align spatial dependencies for DG in MTS data, addressing a critical gap in existing methods. We also revised the paper to clearly present these novel contributions in the abstract, introduction, and methodology sections.
> Thank you again for your insightful comments.
>
> ## **Related Work**:
> We appreciate the reviewer for highlighting related works to ASAM. A key difference between the referenced papers and our work lies in the experimental setting: ASAM is designed for domain generalization, whereas the goal of the referenced papers is to achieve superior performance in domain adaptation settings. In the domain adaptation task, the model is trained with labeled source domain data and part of target domain data, usually without labels or a few label information, to achieve high performance in the target domain. On the other hand, domain generalization evaluates the model on entirely unseen target domains during the training phase.
> Beyond the experimental setting, our contributions share some similarities with those of SEA++, as mentioned by the reviewer. Both approaches aim to address spatial information across domains; however, the methodologies differ significantly. SEA++ aligns sensors based solely on corresponding positions, while our approach dynamically considers correlations across all sensors. This dynamic alignment provides greater generalizability. For instance, as previously mentioned, SEA++ may struggle to generalize between a left-handed subject in one domain and a right-handed subject in another, whereas our approach can effectively align the left hand of one subject with the right hand of another. To further clarify this distinction, we have updated the paper by including a comparison with unsupervised domain adaptation frameworks in Appendix A.2.

---

> > ### Comment · Reviewer_egig · 2024-11-20
> >
> > Thanks to the authors for the detailed rebuttal. I still have the following concerns:
> >
> > 1. for the abstract, despite still being a bit long, it's fine.
> >
> > My major concerns are the novelty and related works.
> >
> > 1. the authors have discussed SEA++. However, relevant discussions of CauDiTS are missing. Actually, the idea of spatial alignment for MTS domain adaptation was first proposed by SASA (AAAI 2021), which should also be discussed and compared.
> >
> > 2. as stated, the major contribution of this work is introducing the idea of spatial alignment into MTS domain generalization. Despite the fact that there exist some technique implementation differences in comparison with SASA, SEA++, CauDiTS, and the proposed work, the high-level idea remains the same, i.e., learning domain-invariant variable-wise dependencies. Therefore, I'm not convinced by the novelty of this work.
> >
> > 3. Furthermore, aligning spatial dependencies for MTS domain adaptation makes sense, i.e., learning shared spatial dependencies between TWO domains to enhance the feature transferability. However, under the domain generalization task setting, are the spatial relationships aligned in the training set necessarily useful for the test domain? What if the test domain has some domain-specific dependencies that are not shared with the training set domain?
> >
> >
> > For the additional experiments:
> >
> > 1. what is the task setting of the Boiler dataset? which domains are chosen for training and testing?  Also, there seems to be no difference in the performance of all the methods. What are the advantages of the proposed method? (Almost all methods achieved an accuracy of 99, which is very surprising.)
> >
> > 2. why choose the SMAP dataset and how do you construct domain generalization task settings for such an anomaly detection dataset?
> >
> > Besides, I've checked the source codes and found they are incomplete and of low quality. There are no instructions for readers to reproduce the experimental results and only include one dataset (SD-Gesture)
> >
> > Considering the aforementioned concerns, I'm more inclined to reject the paper.

---

> > > ### Author Response · Authors · 2024-11-24
> > >
> > > ## **Related works and novelty:**
> > > We acknowledge that our comparison focuses primarily on the SEA++ baseline. However, we believe SEA++ is the most directly comparable work among the three UDA references. This is due to its additional loss function, which aligns two representations influenced by an adjacency matrix capturing spatial dependencies. While SASA and CauDiTS also leverage spatial dependencies in MTS data for adaptation, their methodologies—extracting domain-invariant representations as associative structures (SASA) and disentangling domain-common causal rationales from domain-specific non-causal correlations (CauDiTS)—align more closely with existing work (b), as depicted in Figure 1 of our paper, rather than with our proposed approach. **Additionally, although these methods share similarities with our work, the fundamental experimental settings differ: our work is conducted in a domain generalization (DG) setting, whereas the references operate within a UDA framework.**
> > >
> > >
> > > As the reviewer mentioned in concern 3, aligning spatial dependency with/without a given target feature is severely different. Our model aims to handle situations where the target domain has some domain-specific dependencies that are not shared with the observed source domain, which the reviewer mentioned in major concern 3. By dividing the source domains into two and adopting a spatial dependency alignment module in the second phase, our model is capable of addressing such situations effectively. Figure 1 of our paper clarifies the difference between existing DG works and ours handling the target domain (test domain), conveying different sensor relations with the observed domains (source domains).
> > >
> > >
> > > ## **Task setting of Boiler dataset:**
> > > The detailed experimental setting for the Boiler dataset is stated in the paper on lines 900-908.
> > > The Boiler dataset is composed of three separate domains and is commonly utilized in unsupervised domain adaptation tasks, where the objective is to adapt a single source domain to a single target domain, often yielding near-perfect accuracy. However, since ASAM framework is proposed in a domain generalization (DG) framework, we structured the experiment as follows: the model was trained using two domains and evaluated on a separate, unseen domain. This approach inherently introduces more diverse domain information during training. Notably, the accuracy of all methods, including ASAM, approached 100, suggesting that the Boiler dataset poses relatively low complexity in a DG setting. Despite this, the results reinforce that ASAM is adaptable across diverse domains and can be effectively applied to settings beyond human activity recognition.
> > >
> > >
> > > ## **About SMAP dataset:**
> > > We choose the SMAP dataset to strengthen the difference between existing DG methods and ours. SMAP is an expert-labeled telemetry anomaly data. From the data, we defined the domain that has the same channel ID. Specifically, we utilized the ID that starts with 'A' in the given dataset and selected the ID that has a similar temporal length. From the acquired seven domains, we divided two domains for the first phase data, two distinct domains for the second phase, and the remaining three phases for evaluating ASAM. Each domain is comprised of 25 sensors, and the goal is to detect anomaly sensor signals. The details of the experimental SMAP dataset are written in lines 882-892.
> > >
> > > -------------------
> > >
> > > We additionally uploaded the HHAR dataset and updated the code with a README file to run the HHAR dataset. The other datasets and their commands are planned to be updated when publically available.

---

> > > > ### Comment · Reviewer_egig · 2024-11-26
> > > >
> > > > Thanks to the authors for the rebuttal. I'll keep the discussion with other reviewers.

---

> ### Author Response · Authors · 2024-11-19
> **Response to Reviewer egig [2/3]**
>
> ## **Motivation of Channel decorrelation loss**:
> As mentioned in our paper, channel decorrelation loss reduces the redundant information from the given data. When aligning spatial dependencies between the first-phase and second-phase distributions with the feature correlation matrix $\mathbf{D}$, there are a lot of matrices that can align the two distinct distributions. Among the matrices, we want to find the matrix that alters the least, where the identity matrix is the corresponding least matrix. Therefore, we apply the channel decorrelation loss to regularize the model and reduce the complicated redundant data.
>
> ## **Additional Experiment on new datasets**:
> Based on the reviewer's concern, we additionally experimented with two more datasets, which are Boiler and SMAP. The SleepEDF dataset the reviewer mentioned is a single sensor dataset, which is not appropriate for validating the efficacy of ASAM. The experimental result is shown below, and we select the baselines that ranked higher than second in at least one dataset in our main experiment table (Table 1). The performance metrics for the Boiler and SMAP datasets are accuracy and recall, respectively. From the table, our model marked second in the boiler dataset and first in the SMAP dataset. Surprisingly, existing DG-based methods failed to find anomalies in the SMAP dataset, while ASAM can reasonably find the anomalies. We think this experimental observation is reasonable since existing methods try to extract domain-invariant information, which is highly related to the dominant class, which is the normal value. The details of the experimental setting are provided in Appendix A.1.9.
>
> | Model     | 2SRNN  | SimpleAtt | STCN-GR | GSAT | CAL | DisC | ASAM(Ours)  |
> |-----------|--------|-----------|---------|------|-----|------|-------|
> | Boiler      | 99.67  | 99.09     | 99.73   | $\mathbf{99.93}$    | 92.23   | 92.16    | 99.87 |
> | SMAP      | 67.76  | 70.68     | 67.53   | 0    | 0   | 0    | $\mathbf{70.79}$ |
>
> ## **Experimental setting clarification**
> We acknowledge that the experimental setting might be challenging to interpret.
> We conducted the experiment of comparing the performance with the multi-source domain generalization setting, where several source domains exist for model training. The amount of training data (source domains) across all baselines and ours are identical, ensuring fairness. Besides, ASAM divides the training data into two for the two-phase learning architecture in ours.
> We conduct two types of experiments: cross-subject and cross-repetition. Since the definition of a domain can vary depending on the context of dataset collection, we regarded a repetition and a subject as a domain for the sEMG and HAR datasets, respectively.
>
> ## **Model computation**:
> Thank you for pointing out the need for computational analysis in ASAM. Even though we calculated the execution time of ASAM, which is included in the extra experimental setting section of Appendix A.1.5, it would be more reasonable to compare the execution time of every baseline. The comparison table is shown below. The execution time of ASAM is not the fastest, but we think the result is similar to the baselines. We uploaded the table and updated the paper in the Appendix.
>
> | Model   | ASAM(Ours) | GSAT   | CAL    | GREA   | DisC   | PICCA  | TCN    | 2SRNN  | SimpleATT | STCN-GR | GNN-SD  |
> |-----------|-------------|--------|--------|--------|--------|--------|--------|--------|-----------|---------|---------|
> | UCI-HAR   | 1.69      | 0.67 | 0.89  | 0.91  | 0.89 | 1.07 | 2.86 | 0.32  | 0.28     | 2.59  | 2.77  |
> | HHAR      | 2.94      | 2.88 | 4.04 | 3.97 | 4.01 | 1.52 | 3.51 | 1.21 | 0.32    | 3.42  | 3.55  |

---

> ### Author Response · Authors · 2024-11-19
> **Response to Reviewer egig [3/3]**
>
> ## **Additional Experiment**:
> To strengthen our contribution, We conducted a simple Jensen–Shannon divergence (JS divergence) analysis across domains (subjects) to assess the distribution discrepancy of each domain compared to the others. For each evaluation domain, we calculate the mean JS-divergence value with respect to all other training domains to represent its distributional difference. The table compares the performance of three feature-learning baselines and ASAM on the HHAR dataset. The feature-learning baselines demonstrated strong accuracy on Subject 2, which has the lowest JS-divergence value (0.0868), but struggled significantly on Subject 1, where the distribution is most different (highest JS-divergence value (0.1304)), suggesting that as the distribution gap increases, these methods face limitations in capturing domain-invariant features. In contrast, ASAM achieved a relatively consistent performance across both subjects, with a smaller gap than the baselines. This result validates the need for our method and its insufficiency of feature learning methods (as illustrated in Figure 1-a of our paper) on domains with large distribution gaps with others.
> We further propose that in class-imbalanced scenarios such as anomaly detection classification within a multi-domain context, existing approaches that focus on extracting domain-sharing uniform information (as illustrated in Figure 1-b of our paper) often predominantly capture information from the majority class, neglecting the minority class. In contrast, an "alignment"-based model like ASAM is capable of extracting more meaningful information from the minority class by leveraging the distinct spatial dependencies between normal and abnormal instances. To substantiate this claim, we conducted an additional experiment on the time-series anomaly detection dataset, SMAP. We compare the recall values, which indicate how effectively the models detect anomalies, against three existing DG baselines that achieved at least the second-highest performance in our main experiment (Table 1). The results presented in the table demonstrate that while domain generalization-based models fail to detect abnormal classes effectively, ASAM performs reasonably in identifying anomalies. We also included the details of this experiment in Appendix A.1.7\sim8.
>
>
> |                | Subject 1 | Subject 2 | Subject 3 | Subject 4 | Subject 5 |
> |----------------|-----------|-----------|-----------|-----------|-----------|
> | **JS-divergence** | 0.1304    | 0.0868    | 0.1024    | 0.0978    | 0.0981    |
> | **2SRNN**      | 24.00     | 81.15     | 27.77     | 91.32     | 72.64     |
> | **SimpleAtt**  | 39.40     | 88.80     | 52.63     | 89.55     | 87.84     |
> | **STCN-GR**    | 42.64     | 93.00     | 45.91     | 92.37     | 72.42     |
> | **ASAM**   | 52.03     | 92.44     | 53.26     | 83.70     | 91.74     |
>
>
> |      | GSAT | CAL | DisC | ASAM(Ours)  |
> |-----------|------|-----|------|-------|
> | SMAP      | 0    | 0   | 0    | $\mathbf{70.79}$ |

---

### Official Review · Reviewer_Q3Mk · 2024-11-01

**Soundness:** 3
**Presentation:** 3
**Contribution:** 2
**Rating:** 5
**Confidence:** 2

**Summary:**

This paper presents a model named ASAM to address the domain generalization setting for multivariate time-series data analysis tasks. ASAM addresses these challenges by introducing a DG layer and a loss function to align spatial dependencies across domains. The method utilizes a two-phase approach: first capturing spatiotemporal features and then aligning spatial dependencies using a GNN-based architecture. Then a two-view regularization method is introduced to further capture underlying spatial dependency in both phases and is comprised of spatial decorrelation loss and Gaussian kernel loss. The model is evaluated on four real-world datasets, demonstrating superior performance over recent baselines in tasks like gesture recognition and human activity recognition.

**Strengths:**

S1. The paper introduces ASAM, a model specifically designed to address spatial dependency alignment in domain generalization setting for multivariate time-series data. The proposed model is claimed to be the pioneering work in this topic.

S2. The paper provides a theoretical analysis demonstrating how the DG layer assimilates invariant information, which provides foundation for domain generalization in multivariate time series data.

S3. Experiments show that the proposed model outperforms ten recent baselines across four datasets on two tasks.

**Weaknesses:**

W1. I am not quite convinced by the motivation of the problem statement. Domain generalization is typically critical in scenarios where no data is accessible for target domains. However, in this paper's tested scenarios, including gesture and human activity recognition, datasets seem to be reasonably accessible. I doubt whether domain generalization use case really applies to these two tasks.

W2. While the proposed model is claimed to address domain generalization for multivariate time series data. The evaluation is conducted on datasets which mainly focused on human activity and gesture recognition tasks. Broader validation on a wider variety of domains could strengthen the generalizability claims of the framework. For example, multivariate time series from traffic sensors also exhibit spatial dependencies.

W3. For the source domain, it is not clear how the spatial dependency graph is constructed for the source domain data. If it is a KNN graph, it seems meaningless to derive the spatial dependencies for the target domain. We just need to use this rule to derive the graph for the target domain.

W4. The impact of the spatial and temporal modeling layers on the proposed model is not sufficiently explored. The paper uses basic GNN + LSTM architectures, but it would not clear how the model performs different variants of spatiotemporal models, such as more advanced models for human activity recognition.

W5. There seems to be a conflict between the function of the spatial layer and the channel decorrelation loss. The spatial layer employs message passing among different nodes (sensors) according to the spatial dependency graph. However, the representations, which encode the spatial correlations, are processed to approximate identity matrix after inner product. How can the inner product produce identity matrices for entries that are already produced by considering spatial dependencies?

**Questions:**

Please clarify the comments in W1-W5.

---

> ### Author Response · Authors · 2024-11-19
> **Response to Reviewer Q3Mk [1/2]**
>
> We sincerely appreciate the reviewer’s insightful feedback and have made every effort to address the identified concerns and weaknesses in a thorough and thoughtful manner.
>
> ## **Motivation and new dataset**:
> Both the human activity recognition and hand gesture recognition tasks are utilized in the supervised learning or domain adaptation field, which seems appropriate to point out the accessibility of data the reviewer mentioned. However, besides our work, there are some other works that address domain generalization in human activity recognition~[1,2]. Additionally, while there are considerations regarding domain accessibility due to confidentiality, the DG setting aims to achieve good performance on new domains, such as unseen subjects or new repetitions from specific subjects. In this context, applying DG to HAR and sEMG datasets under the paper’s scenario can be regarded as reasonable. Furthermore, to relieve the reviewer's concern, we validate the effectiveness of ASAM by adding two more datasets, Boiler and SMAP. Especially, we believe that the time series anomaly detection dataset, SMAP, is a dataset that is not readily accessible since real-time estimation is crucial. The experimental result is shown below, and we select the baselines that ranked higher than second in at least one dataset in our main experiment table (Table 1). The performance metrics for the Boiler and SMAP datasets are accuracy and recall, respectively. From the table, our model marked second in the Boiler dataset and first in the SMAP dataset. Surprisingly, existing DG-based methods failed to find anomalies in the SMAP dataset, while ASAM can reasonably find the anomalies. We think this experimental observation is reasonable since existing methods try to extract domain-invariant information, which is highly related to the dominant class, the normal value. Moreover, since the traffic sensor dataset is usually applied to predict future values similar to the regression task, we could not find the appropriate traffic sensor dataset that is available in our setting, which is the classification setting.
>
>
> |      | 2SRNN  | SimpleAtt | STCN-GR | GSAT | CAL | DisC | ASAM(Ours)  |
> |-----------|--------|-----------|---------|------|-----|------|-------|
> | Boiler      | 99.67  | 99.09     | 99.73   | $\mathbf{99.93}$    | 92.23   | 92.16    | 99.87 |
> | SMAP      | 67.76  | 70.68     | 67.53   | 0    | 0   | 0    | $\mathbf{70.79}$ |
>
>
> [1] Domain Generalization for Activity Recognition via Adaptive Feature Fusion - ACM Trans. Intell. Syst. Technol. 14(1)
> [2]CrossHAR: Generalizing Cross-dataset Human Activity Recognition via Hierarchical Self-Supervised Pretraining. Proc. ACM Interact - Mob. Wearable Ubiquitous Technol. 8(2)
>
> ## **Clarification of spatial dependency graph**:
> The spatial dependency we claimed is related to the domain generalization layer, in which only the second phase data is processed. Our objective is to align the second-phase data such that its distribution becomes similar to that of the first-phase data. Consequently, the first phase data has no spatial dependency graph. However, within the spatial layer in our spatiotemporal model $f\_{\theta}$, both the first phase and second phase are associated with their respective spatial graphs. We employ a GCN with a learnable adjacency matrix, which is not a KNN graph. We acknowledge that the lack of a detailed explanation of our spatiotemporal model may have contributed to readability issues, and we sincerely apologize for any confusion caused. We updated the details of our GNN-LSTM-based spatiotemporal model in Appendix A.1.4.

---

> ### Author Response · Authors · 2024-11-19
> **Response to Reviewer Q3Mk [2/2]**
>
> ## **Impact of spatial and temporal layers**:
> Thanks for pointing out an interesting question. Even though our paper aims to accentuate the need for spatial alignment through the domain generalization layer, the overall performance is highly related to the structure of our spatiotemporal model. We additionally modified the spatial structure to CNN and an attention-based model, which closely resembles the recent HAR-based models~[3,4]. The performance variation is validated through two datasets, SD-gesture and UCI-HAR, to align with our temporal variation experiment in Table 3-(c). The table below shows the performance variation, and we conclude that the simple GNN-LSTM-based model we adopt shows the highest performance under these variations. We updated the paper by incorporating the table ablating the spatial layer in Table 3.
>
> | Dataset     | SD-gesture  | UCI-HAR |
> |-----------|--------|-----------|
> | CNN      | 88.13  | 82.45     |
> | Attention      | 83.29  | 86.09     |
> | GNN (ASAM)      | 93.04  | 92.42     |
>
>
> [3] Human Activity Recognition using 1D-CNN and Stacked LSTM - International Conference on Signal and Information Processing (IConSIP) 2022
> [4] Spatiotemporal Attention-Based Framework for Human Activity Recognition. Applied Computational Intelligence and Soft Computing, 2024
>
> ## **Conflict between losses**:
> As the reviewer mentioned, applying both domain generalization loss and channel decorrelation loss makes it seem like they have a conflict. However, in the point of information bottleneck, which is related to our theoretical analysis, the domain generalization loss is trying to minimize the conditional mutual information $I(\mathbf{Z}\_{\text{dg}}, \text{DG} | \mathbf{X})$, while both the domain generalization loss and channel decorrelation loss are related to maximizing the mutual information between the learned representation and the input, $I(\mathbf{Z}\_{\text{dg}}, \mathbf{X})$. Therefore, even though the two losses may seem to be in conflict, the employment of both losses indicates extracting all the essential information for adapting to a new domain and discarding the redundant data that is related to the given source data $X$. Consequently, our learned representation with the channel decorrelation loss optimizes the MTS sensor-based information bottleneck representation, which has no redundant data. Therefore, we conclude that the domain generalization loss incorporated with the channel decorrelation loss will lead to the information bottleneck of MTS sensor data.
>
> Overemphasizing $I(\mathbf{Z}\_{\text{dg}}, \mathbf{X})$ can result in trivial representations, negatively impacting overall model performance. To investigate this, we evaluated ASAM on the UCI-HAR dataset by varying the influence of the channel decorrelation loss, controlled by $\lambda\_2$. The results demonstrate that as $\lambda\_2$ increases, performance degrades. This trend suggests that excessive focus on maximizing $I(\mathbf{Z}\_{\text{dg}}, \mathbf{X})$ can lead to suboptimal representations, ultimately hindering the model's effectiveness.
>
>
>
> | $\lambda\_2$     | 0  | 1e-03  | 1e+00 | 1e+02 | 1e+04
> |-----------|---------|----------|----------|---------|---------|
> | UCI-HAR   | 89.98  | 93.02   | 91.81   | 88.37  |  76.61 |

---

### Official Review · Reviewer_7vYk · 2024-11-02

**Soundness:** 3
**Presentation:** 3
**Contribution:** 3
**Rating:** 6
**Confidence:** 4

**Summary:**

The authors proposed a simple but interesting framework to incorporate spatial information into domain generalization. To align the spatial information between different domains, the authors split the domains as two parts. A model is trained with one part and further tuned by another part. The input-aware graph generation is designed for the tuning process.

**Strengths:**

1.	To align the spatial information between different domains, the authors split the domains as two parts. A model is trained with one part and further tuned by another part. The input-aware graph generation is designed for the tuning process. Though it can be further improved, the idea is interesting.

2.	Experiments are comprehensive.

**Weaknesses:**

1.	The definition of spatial information is unclear. In this work, the authors align the sensor features between domains by reducing their distances, so it seems that the authors treat the node features as the spatial information given they are updated by propagation. However, in my opinion, the spatial information should include both the sensor features and the spatial structure between these sensors. In this method, the spatial structure, i.e., their edges, cannot be aligned explicitly.

2.	Fig 1 is not informative. What do you mean by a, b, c, d and A, B, C, D. How do you argue 'ours' is better than (a) and (b). These are all not clear.

3.	Regarding Eq. (5), I dont understand why reducing the loss can alleviate the redundant data. D = ZZ^T should be like the correlations between sensors. I think it would remove the correlation information  when reducing the loss. Can the authors explain this?

4.	Relevant work is not discussed [1], which also transfer spatial information during alignment, though it was designed for UDA.

5.	Is there any rule to make better split for phase 1 and 2?


[1] Wang Y, Xu Y, Yang J, et al. Sea++: Multi-graph-based higher-order sensor alignment for multivariate time-series unsupervised domain adaptation[J]. IEEE transactions on pattern analysis and machine intelligence, 2024.

**Questions:**

See abve

---

> ### Author Response · Authors · 2024-11-19
> **Response to Reviewer 7vYk [1/3]**
>
> We sincerely appreciate the reviewer’s insightful feedback and have made every effort to address the identified concerns and weaknesses in a thorough and thoughtful manner.
>
>
>
> ## **Definition of spatial information**:
> We apologize for any confusion caused regarding the definition of spatial information. The spatial information referenced by the reviewer pertains to Equation 1 in our paper, which defines the learnable adjacency matrix to transform the given distribution of the second phase data into the distribution of the first phase data, and the node features of $\mathbf{Z}\_{\text{dg}}$ is the resultant transformed features reflected by the spatial information.
>
> While we acknowledge the reviewer’s perspective that spatial information should include both the node features and the spatial structure (i.e., the edges between sensors), our approach focuses on aligning the spatial dependencies encoded in the adjacency matrix, rather than explicitly aligning the spatial structure itself. We hypothesize that when these spatial dependencies are effectively aligned, the distributions of $\mathbf{Z}\_{\text{ori}}$ and $\mathbf{Z}\_{\text{dg}}$ will naturally become similar, achieving our objective.
>
> We also recognize that the adjacency matrix in our method is derived solely from the second-phase data and, as such, does not have a direct reference for explicit edge alignment. This limitation prevents us from aligning the spatial structure explicitly in the current framework. We appreciate the reviewer pointing out this aspect and acknowledge that incorporating explicit edge alignment into the learnable adjacency matrix is a promising direction for future work.

---

> ### Author Response · Authors · 2024-11-19
> **Response to Reviewer 7vYk [2/3]**
>
> ## **Clarification of Figure 1 and motivation**:
>
> We thank the reviewer for pointing out the lack of clarity in Figure 1 and apologize for any confusion caused. In the updated paper, we have revised the figure and included detailed explanations of the notations. Specifically, the lowercase letters (a, b, c, d) represent learned sensor features influenced by domain-specific information, while the uppercase letters (A, B, C, D) represent features aligned with domain-generalized information. This clarification has been added to the updated paper.
>
> ### Strength of ASAM compared to existing methods.
>
> Figure 1 illustrates the strength of our method, particularly in scenarios where target domains possess significantly different spatial structures compared to source domains. By leveraging our DG layer as an adaptive sensor dependency alignment module, ASAM effectively addresses challenges that existing approaches fail to handle, as illustrated in the T1 domain in the figure. Below, we summarize key insights and provide additional experiments to clarify why ASAM outperforms existing methods (a) and (b):
>
> - Large domain gaps:
>
> Existing feature-learning methods, represented by (a) in Figure 1, perform poorly when there are substantial distribution discrepancies between training and evaluation domains. To demonstrate this, we computed the Jensen–Shannon (JS) divergence between domains (subjects) in the HHAR dataset, which quantifies the distributional difference of each domain compared to others. For each evaluation subject (i.e., domain), we compute the average JS-divergence or average accuracy over all other subjects.
> The table below shows that feature-learning baselines achieve strong accuracy on Subject 2 (lowest JS-divergence: 0.0868) but struggle on Subject 1 (highest JS-divergence: 0.1304), highlighting their limitations in capturing domain-invariant features. In contrast, ASAM maintains consistent performance across all domains, with a much smaller performance gap, validating its ability to generalize effectively even with large domain gaps.
>
>
>
> |                | Subject 1 | Subject 2 | Subject 3 | Subject 4 | Subject 5 |
> |----------------|-----------|-----------|-----------|-----------|-----------|
> | **JS-divergence** | 0.1304    | 0.0868    | 0.1024    | 0.0978    | 0.0981    |
> | **2SRNN**      | 24.00     | 81.15     | 27.77     | 91.32     | 72.64     |
> | **SimpleAtt**  | 39.40     | 88.80     | 52.63     | 89.55     | 87.84     |
> | **STCN-GR**    | 42.64     | 93.00     | 45.91     | 92.37     | 72.42     |
> | **ASAM**   | 52.03     | 92.44     | 53.26     | 83.70     | 91.74     |
>
>
> - Class imbalance:
>
> In class-imbalanced scenarios, such as anomaly detection, existing DG methods often focus on extracting shared information across domains, predominantly capturing features from majority classes and neglecting minority ones. ASAM overcomes this limitation by leveraging alignment to extract meaningful spatial dependencies that differentiate normal and abnormal instances.
> To validate this, we conducted experiments on the SMAP anomaly detection dataset, comparing ASAM with three DG baselines (GSAT, CAL, and DisC). The recall values below show that while existing methods fail to detect anomalies effectively, ASAM achieves significantly better performance:
>
>
> |      | GSAT | CAL | DisC | ASAM(Ours)  |
> |-----------|------|-----|------|-------|
> | SMAP      | 0    | 0   | 0    | $\mathbf{70.79}$ |
>
> We also included the details of those experimental results in Appendix A.1.7~8.

---

> ### Author Response · Authors · 2024-11-19
> **Response to Reviewer 7vYk [3/3]**
>
> ## **About Channel Decorrelation Loss**:
> We aimed to utilize the term "redundant" as the complicated spatial dependency when there are simple dependencies in which the resultant representation is similar. The identity matrix could represent the simplest dependency, which leads to the channel decorrelation loss formulated like equation 4 in our paper.
> By encouraging $\mathbf{D}$ to resemble the identity matrix, this loss function seeks to capture the simplest and most direct correlations while avoiding the over-propagation of information across sensors. Therefore, we add this channel decorrelation loss as a regularizer to prevent the model from learning too complicated dependencies. Furthermore, our theoretical analysis states that when employing both the domain generalization loss and channel decorrelation loss, our representation satisfies the information bottleneck for the MTS sensor, which implies that the representation eliminates the redundancy within the given source data. However, as the reviewer mentioned, solely applying the channel decorrelation loss will lead to reducing the correlation between sensors, which seems irrelevant in eliminating the redundant data. Therefore, we conduct an ablation study about the effect of channel decorrelation by adjusting $\lambda\_2$ in the UCI-HAR dataset, and its result is shown below. From the result, when $\lambda\_2$ becomes higher, the performance deteriorates, which can be interpreted as not eliminating the redundant data.
> Thanks for pointing this out, and we apologize for the confusion related to the term "redundant". We clarified this point in our updated paper.
>
> | $\lambda\_2$     | 0 | 1e-03  | 1e-00 | 1e+02 | 1e+04
> |-----------|--------|--------|-----------|---------|---------|
> | UCI-HAR   | 89.98 | 93.02  | 91.81   | 88.37  |  76.61 |
>
> ## **Related work**:
> As noted by the reviewer, we excluded comparisons with domain adaptation methods to avoid potential confusion. Domain adaptation typically partitions the data into a source domain and a target domain, optimizing the model using the entire source dataset and partial target data to improve performance on the remaining target data. In contrast, domain generalization aims to achieve high performance on domains that the model has not encountered during training.
>
> Additionally, the related work mentioned by the reviewer shares some similarities with our approach, as it also considers spatial relations. However, their method compares spatial relations between source and target domains based on identical indices. This assumption raises concerns in cases such as a left-handed user in the source domain and a right-handed user in the target domain. In such scenarios, it would be more effective to match the two domains symmetrically across multiple sensors rather than relying on fixed sensor positions. By adopting an adaptive spatial alignment module, our work can symmetrically align the sensor dependencies, addressing the potential limitations of the related work. A section introducing the unsupervised domain adaptation methodology and comparing the related work mentioned by the reviewer with ASAM is included in Appendix A.2.
>
> ## **Data split**:
> To figure out if there is any better split method than random split, we performed every scenario on the first subject in the Ninapro DB 5 dataset. Since the dataset is comprised of 6 domains and we applied 2 domains for the first phase and 2 for the second phase, there are six possible scenarios for the first phase domains while fixing the evaluation domains as 5th and 6th repetition, [(0,1), (0,2), (0,3), (1,2), (1,3), (2,3)]. The experimental result is shown below. The experimental result validates that the difference is not significant.
>
> | First phase domains   | 0,1   | 0,2   | 0,3   | 1,2   | 1,3   | 2,3   |
> |--------|-------|-------|-------|-------|-------|-------|
> | Acc.  | 86.91 | 86.90 | 86.87 | 85.77 | 84.72 | 85.21 |

---

> > ### Comment · Reviewer_7vYk · 2024-11-28
> >
> > I appreciate the authors' detailed response, which has partially addressed my concern. However, I think the difference of the related work and this work should be discussed in the main manuscript. The related work highly shared similarities with this work, they all discuss the spatial information and all for addressing the domain gaps, although one for DG and another for DA. Emphasizing the difference can help to highlight the unique contribution of this work.

---

> > > ### Author Response · Authors · 2024-11-28
> > >
> > > Thank you for your valuable feedback and thoughtful suggestions. We appreciate your emphasis on discussing the distinction between our work and related research. To address this, we have included a discussion on DA methods and their differences from our approach in the main manuscript.
> > >
> > > This addition helps clarify our work's unique contributions and situate them more effectively within the broader research landscape. We would be delighted to discuss any further suggestions or insights.

---

### Official Review · Reviewer_X3of · 2024-11-04

**Soundness:** 3
**Presentation:** 2
**Contribution:** 3
**Rating:** 5
**Confidence:** 2

**Summary:**

This paper introduces a framework called ASAM that enhances the generalization of multivariate time-series sensor data across different domains by adaptively aligning spatial dependencies. ASAM employs a two-phase approach, including a domain generalization layer and loss functions designed for spatial alignment, to capture and align spatial dependencies across domains. Experimental results demonstrate that ASAM outperforms several recent baseline models on multiple real-world datasets, showing its effectiveness in domain generalization tasks.

**Strengths:**

**Innovation and Originality:** This paper focuses on the domain generalization problem in multivariate time series sensor data, particularly addressing spatial dependency alignment. This is a noteworthy research topic.

**Theoretical Analysis:** This paper provide a theoretical analysis of how the domain generalization layer and loss functions promote domain generalization, which enhances the credibility of the approach and offers a theoretical foundation for domain generalization.

**Experimental Design and Results:** The experimental section covers multiple real-world datasets and compares the method against several baseline models, demonstrating the effectiveness and superior performance of ASAM across various datasets.

**Code and Resources:**  A link to the code is provided, which aids the community in reproducing and further exploring this work.

**Weaknesses:**

**Weaknesses:**

1. **Motivation and Insight:** The concept of "alignment" as presented seems somewhat unconvincing. Given the heterogeneity of time series data, can spatial dependencies across different domains truly be aligned? What specific issues does the alignment process aim to address? Does it imply an assumption that the source domain should contain distributions similar to those in the target domain?

2. **Writing Clarity:** While the proposed method is theoretically innovative, certain sections, especially regarding the algorithmic implementation details and explanations of mathematical formulas, could be more clear and detailed. Symbols should be well-defined upon first appearance. For instance, in Section 3.2, what does $F$ represent in $C \in \mathbb{R}^{F \times T}$? Additionally, why do the so-called representations $Z \in \mathbb{R}^{N \times T}$ share the same dimensions as the input data $X$? Typically, representations exist in a hidden space of dimension $D$. The appearance of subscripts and superscripts should also be explicitly explained to clarify their role in the workflow. Such issues make the complex algorithm harder to understand. Significant improvements in the writing of the methods section are necessary.

3. **Additional Concerns:**
   - Is the so-called DG layer merely a simple GCN? As presented, it is unclear how they differ.
   - Furthermore, does simultaneously optimizing both the graph structure and the GNN lead to potential bi-level optimization issues? [1][2][3]
   - The authors could consider conducting additional generalization experiments or discussions to more objectively evaluate the "alignment" strategy and better position this work. For example, under what circumstances (e.g., when there is a substantial gap between the source and target domains) might the alignment approach become ineffective?

[1] Learning Discrete Structures for Graph Neural Networks
[2] Discrete Graph Structure Learning for Forecasting Multiple Time Series
[3] Pre-training Enhanced Spatial-temporal Graph Neural Network for Multivariate Time Series Forecasting

**Questions:**

See weakness.

---

> ### Author Response · Authors · 2024-11-19
> **Response to Reviewer X3of [1/2]**
>
> We sincerely appreciate the reviewer’s insightful feedback and have made every effort to address the identified concerns and weaknesses in a thorough and thoughtful manner.
>
> ## **Motivation and Insight**:
>
> We appreciate the reviewer’s feedback and acknowledge that the term "alignment" might have caused some misunderstanding. To clarify, in this paper, "alignment" refers specifically to matching the relative spatial dependencies—i.e., the relationships or correlations between sensors—across domains, rather than directly matching their raw sensor signals or distributions. This distinction is critical to understanding our approach.
>
> The concept of alignment in our method addresses the challenge of domain shifts caused by differences in spatial dependency structures, which can arise from variations in the physical setup, sensor configuration, or even contextual conditions across domains. These shifts in spatial relationships, rather than misalignments of raw signals, are the focus of our framework. By leveraging the proposed domain generalization (DG) loss, we optimize the model to align these relative dependencies, enabling robustness across diverse domains.
>
> As illustrated in Figure 3 of the paper, our alignment process does not assume that the source and target domains share similar distributions or raw signals. Instead, it enforces the learning of invariant spatial relationships that generalize well to unseen domains. This approach is key to handling the heterogeneity of multivariate time-series (MTS) data, as further detailed in our experiments.
>
> We hope this clarification addresses the reviewer’s concern, and we believe our approach offers a robust and novel solution to the challenges posed by heterogeneous spatial dependencies in MTS data.
>
>
> ## **Writing Clarity**:
>
> ### Clarification of Variables and Notations:
> - We have explicitly defined $F$ in $\mathbf{C} \in \mathbb{R}^{F \times T}$ as the feature dimension for each temporal dimension $T$.
> - Additional descriptions have been included for subscripts and superscripts to enhance readability and understanding of the mathematical formulas and their roles in the workflow.
> - Algorithm 1 and its corresponding explanation have been revised for greater clarity.
>
> ### Dimensionality of Representations:
> - We set $T$ to represent the temporal dimension, ensuring it aligns with the input $\mathbf{X}$ due to the use of a residual connection in our GNN-LSTM-based spatio-temporal model. This residual connection inherently maintains the same temporal dimension $T$ between the input $\mathbf{X} \in \mathbb{R}^{N \times T}$ and the learned representations $\mathbf{Z} \in \mathbb{R}^{N \times T}$.
> - Without the residual connection, the temporal dimension $T$ of $\mathbf{Z}$ can be freely adjusted, as noted by the reviewer. To explore this flexibility, we conducted an additional ablation study on the UCI-HAR dataset, varying $T$ while removing the residual connection. The results, summarized below, demonstrate robust performance across different $T$ values. However, the highest performance was achieved by maintaining the same dimension as the input $T$ with the residual connection, which we adopt in our final implementation.
>
>
> | Dimension of $T$     | 64  | 128 (original dimension of $T$) | 256 | 128 + residual|
> |-----------|--------|-----------|---------|---------|
> | Accuracy on UCI-HAR | 86.30  | 86.21     | 88.34   | 92.42|
>
> ### Comprehensive Explanation:
> - To further aid understanding, we have included a detailed explanation of the GNN-LSTM-based spatiotemporal model in the updated manuscript, with additional details provided in Appendix A.1.4.

---

> ### Author Response · Authors · 2024-11-19
> **Response to Reviewer X3of [2/2]**
>
> ## **Additional Concerns**:
>
> 1. Relation between DG layer and Simple GCN: Our DG layer employs a simple GCN with an additional learnable adjacency matrix. The propagation strategy from the GCN model with the learnable adjacency matrix facilitates the alignment of distinct domains.
>
> 2. Bi-level Optimization: The training process of the second phase can be interpreted as a bi-level optimization and its equation can be expressed as $\min\_{\mathbf{A}} \mathbb{E}\_{\mathbf{A}}[f\_{\theta}(\mathbf{A}, \mathbf{X})] \text{s.t.} \min\_{\theta}[\mathcal{L}\_{\text{P2}}]$, where $\mathcal{L}\_{\text{P2}}$ is related with both $\theta$ and $\mathbf{A}$. One of the issues in bi-level optimization is instability due to multi-objectives. It is known that pretraining the inner objective and setting the parameters as the pre-trained weight in advance is a straightforward way to stabilize the learning[1]. Since the first phase can be interpreted as pretraining the spatiotemporal model $f\_{\theta}$, our model can indirectly alleviate the bi-level optimization issues.
>
> 3. Additional generalization experiment:
>
> We thank the reviewer for their suggestion to further evaluate the “alignment” strategy and its potential limitations. To address this, we conducted two additional experimental studies that emphasize the limitations of existing feature learning and graph-based domain generalization (DG) baselines, particularly under challenging conditions such as large domain gaps and class imbalance. These studies objectively highlight the necessity of the proposed “alignment” approach in MTS sensor-based data.
>
> ### Evaluation on substantial domain gaps
>
> To evaluate performance under varying domain gaps, we calculated the Jensen–Shannon (JS) divergence between domains on the HHAR dataset, where each domain represents a subject. For each evaluation subject (i.e., domain), we compute the average JS-divergence or average accuracy over all other subjects. The table below compares the performance of three feature-learning baselines and ASAM on the HHAR dataset. Our results show that feature-learning baselines perform well on Subject 2, which exhibits the smallest JS divergence (0.0868), but suffer a significant drop in accuracy on Subject 1, which has the largest JS divergence (0.1304). This indicates their inability to capture domain-invariant features as the distribution gap increases. In contrast, ASAM demonstrates relatively stable performance across all subjects, with a much smaller performance gap, as shown below:
>
>
> |                | Subject 1 | Subject 2 | Subject 3 | Subject 4 | Subject 5 |
> |----------------|-----------|-----------|-----------|-----------|-----------|
> | **JS-divergence** | 0.1304    | 0.0868    | 0.1024    | 0.0978    | 0.0981    |
> | **2SRNN**      | 24.00     | 81.15     | 27.77     | 91.32     | 72.64     |
> | **SimpleAtt**  | 39.40     | 88.80     | 52.63     | 89.55     | 87.84     |
> | **STCN-GR**    | 42.64     | 93.00     | 45.91     | 92.37     | 72.42     |
> | **ASAM**   | 52.03     | 92.44     | 53.26     | 83.70     | 91.74     |
>
> This experiment validates the effectiveness of our alignment strategy in handling substantial domain gaps, where feature-learning methods fail to generalize effectively.
>
> ### Evaluation on class imbalance
>
> In class-imbalance scenarios, such as anomaly detection, existing DG approaches often fail to capture meaningful domain-invariant information for minority classes, instead focusing on majority class features. ASAM addresses this limitation by leveraging its alignment strategy to extract meaningful spatial dependencies that differentiate normal and abnormal instances.
>
> We conducted additional experiments on the SMAP anomaly detection dataset, comparing ASAM with three DG baselines (GSAT, CAL, and DisC) that achieved strong performance in our main experiments. The results, summarized below, highlight ASAM’s superior ability to identify anomalies, as reflected by its recall score:
>
>
> |      | GSAT | CAL | DisC | ASAM(Ours)  |
> |-----------|------|-----|------|-------|
> | SMAP      | 0    | 0   | 0    | $\mathbf{70.79}$ |
>
> We also included the details of those experimental results in Appendix A.1.7~8.
>
>
>
> [1] Model-Agnostic Meta-Learning for Fast Adaptation of Deep Networks - ICML 2017

---

> ### Comment · Reviewer_X3of · 2024-11-25
> **Thanks for the rebuttal.**
>
> Thank you to the authors for their rebuttal. I remain unconvinced about the explanation regarding "alignment of the relative spatial dependencies." In my view, these spatial dependencies, particularly in a graph structure learning scenario, primarily reflect the relationships between signals from different sensors. As a result, I will maintain my score, albeit with lower confidence.

---

### Comment · Area_Chair_f2zq · 2024-11-26
**Encouragement to Actively Participate in the Discussion Phase**

Dear Reviewers,

Thank you for your valuable contributions to the review process so far. As we enter the discussion phase, I encourage you to actively engage with the authors and your fellow reviewers. This is a critical opportunity to clarify any open questions, address potential misunderstandings, and ensure that all perspectives are thoroughly considered.

Your thoughtful input during this stage is greatly appreciated and is essential for maintaining the rigor and fairness of the review process.

Thank you for your efforts and dedication.

---

### Meta-Review · Area_Chair_f2zq · 2024-12-19

**Metareview:**

(a) Summary of Scientific Claims and Findings
The paper introduces ASAM (Adaptive Spatial Dependency Alignment Model) to address domain generalization (DG) in multivariate time-series (MTS) sensor data. The method focuses on aligning spatial dependencies between domains using a two-phase approach:
A domain generalization (DG) layer that models spatial correlations adaptively.
Two-view regularization losses (spatial decorrelation loss and Gaussian kernel loss) to capture spatiotemporal information while reducing overfitting.
The paper claims that ASAM:
Aligns spatial dependencies across domains without assuming invariant spatial structures.
Incorporates theoretical grounding based on the information bottleneck principle.
Outperforms ten recent baselines across four real-world datasets (e.g., HHAR, SD-Gesture) in classification tasks.

(b) Strengths of the Paper
Innovative Framework: ASAM’s focus on spatial dependency alignment for domain generalization is novel within the context of MTS data.
Comprehensive Evaluation: The method is tested on multiple datasets, demonstrating improved performance compared to baseline models.
Theoretical Analysis: The theoretical connection to the information bottleneck principle enhances the scientific rigor.
Practical Contribution: The authors provide accessible code for reproducing results, which supports transparency and reproducibility.

(c) Weaknesses of the Paper
Limited Novelty: The high-level concept of spatial dependency alignment is similar to existing works in domain adaptation (e.g., SEA++, CauDiTS). The novelty lies primarily in adapting this idea to domain generalization, which is not convincingly differentiated.
The use of basic architectures (e.g., GNN-LSTM) further reduces the perceived technical contribution.

Evaluation Gaps: The datasets primarily focus on human activity and gesture recognition, limiting generalizability claims. Broader datasets, such as traffic or financial time-series, are missing.
The added datasets during rebuttal (Boiler, SMAP) are poorly justified, and the evaluation does not provide meaningful insights (e.g., near-identical performance across methods in Boiler).

Unresolved Methodological Concerns: The effectiveness of spatial alignment for unseen domains is questionable, especially when target domains exhibit domain-specific dependencies not present in training data.
Conflicts between the spatial layer and channel decorrelation loss remain inadequately explained.

Presentation Issues: The clarity of key concepts (e.g., spatial alignment) and experimental settings is lacking.
Missing or incomplete baselines (e.g., SASA, SEA++, CauDiTS) weaken the comparative analysis.

(d) Reasons for Rejection
Insufficient Novelty: The high-level idea of spatial dependency alignment is not sufficiently distinct from existing methods in domain adaptation, which diminishes the contribution to domain generalization.
Evaluation Limitations: The focus on narrow application domains and weak justification for added datasets undermine the generalizability of results. Missing relevant benchmarks further weaken the empirical validation.
Methodological and Theoretical Gaps: The unresolved questions about the effectiveness of alignment for unseen domains and conflicts between loss functions suggest conceptual weaknesses.
Incomplete Rebuttal: Despite detailed responses, key concerns (e.g., novelty, evaluation breadth, methodological clarity) remain unaddressed, leaving significant doubts about the work’s impact.

While ASAM introduces an approach to domain generalization for MTS data, its contributions are not sufficiently robust or novel to meet the high standards of ICLR.

**Additional Comments On Reviewer Discussion:**

Concern: Several reviewers highlighted the similarity between ASAM and existing domain adaptation methods (e.g., SEA++, CauDiTS, SASA). They questioned the novelty of applying spatial dependency alignment to domain generalization.
Author Response: The authors argued that ASAM introduces a novel alignment mechanism for domain generalization, distinct from domain adaptation frameworks. They emphasized ASAM’s dynamic alignment of sensor dependencies, which avoids fixed positional assumptions. However, direct comparisons with SASA and CauDiTS were not provided.
Evaluation: The response partially addressed the concern but failed to demonstrate a significant departure from prior works. Reviewers remained unconvinced about the novelty of the high-level idea.
Effectiveness of Spatial Alignment for Unseen Domains

Concern: Reviewers questioned whether aligning spatial dependencies during training would benefit domains with unseen and unique dependencies.
Author Response: The authors introduced additional experiments to validate ASAM’s robustness under domain gaps (e.g., large Jensen–Shannon divergence across domains in HHAR). They claimed ASAM showed consistent performance compared to baselines.
Evaluation: While the new experiments were appreciated, reviewers noted that they did not conclusively address whether alignment assumptions held for truly unseen domain-specific dependencies.
Evaluation Scope and Dataset Justification

Concern: The reviewers criticized the narrow focus on human activity and gesture recognition datasets, questioning the generalizability of the results. The addition of Boiler and SMAP datasets during rebuttal was seen as weakly justified.
Author Response: The authors explained the domain selection process and the relevance of Boiler and SMAP. They noted that broader datasets (e.g., traffic sensors) were unavailable in their classification setting.
Evaluation: The justification was unconvincing, especially given the near-identical performance of methods on Boiler and the questionable domain split for SMAP. Reviewers maintained that the evaluation lacked breadth.
Theoretical Gaps and Loss Function Conflicts

The rebuttal period demonstrated the authors’ commitment to addressing concerns, but critical issues remained unresolved:
Limited Novelty: The work’s overlap with existing methods (e.g., SEA++, SASA) was not convincingly addressed, and the lack of direct comparisons weakened claims of originality.
Evaluation Gaps: The narrow dataset scope and weak justification for added datasets during rebuttal undermined generalizability claims.
Theoretical and Methodological Issues: Concerns about alignment assumptions, loss function conflicts, and theoretical contributions persisted despite additional experiments and explanations.
Reviewer Consensus: Most reviewers retained their initial concerns and scores, noting that while the paper had potential, it fell short of the standards required for ICLR.

---

### Decision · Program_Chairs · 2025-01-22

Reject